# DO BRAINS AND LLMS PROCESS ALIKE? EXPLORING NEURAL AND MODEL TRAJECTORIES SIMILARITY

## ABSTRACT

Understanding the similarity between large language models (LLMs) and human brain activity is crucial for advancing both AI and cognitive neuroscience. In this study, we provide a multilinguistic, large-scale assessment of this similarity by systematically comparing 16 publicly available pretrained LLMs with human brain responses during natural language processing tasks in both English and Chinese. Specifically, we use ridge regression to assess the representational similarity between LLM embeddings and electroencephalography (EEG) signals, and analyze the similarity between the "neural trajectory" and the "LLM latent trajectory." This method captures key dynamic patterns, such as magnitude, angle, uncertainty, and confidence. Our findings highlight both similarities and crucial differences in processing strategies: (1) We show that middle-to-high layers of LLMs are central to semantic integration and correspond to the N400 component observed in EEG; (2) The brain exhibits continuous and iterative processing during reading, whereas LLMs often show discrete, stage–end bursts of activity, which suggests a stark contrast in their real-time semantic processing dynamics. This study could offer new insights into LLMs and neural processing, and also establish a critical framework for future investigations into the alignment between artificial intelligence and biological intelligence. The code is available at https://anonymous.4open.science/r/57DF.

## 1 INTRODUCTION

The development of large language models (LLMs) has transformed natural language processing (NLP), enabling machines to generate human-like text and perform various linguistic tasks with impressive accuracy (Zhang et al., 2025; Lee et al., 2025; Zhang et al., 2024; Steyvers et al., 2025). However, the mechanisms by which LLMs process and understand language remain largely opaque (Takahashi et al., 2024; Ferraris et al., 2025; Chandrasekharan & Jacob, 2025). This has spurred interest in comparing LLMs to human cognition, particularly regarding how both systems represent and process language. While LLMs excel at language tasks, the extent to which they simulate human cognitive processes is still an open research question.

Studies of AI-human similarity have traditionally focused on behavioral outcomes, comparing AI performance with human data across tasks such as essay writing, image recognition, and logical reasoning (Ashktorab et al., 2021; Kumar et al.,

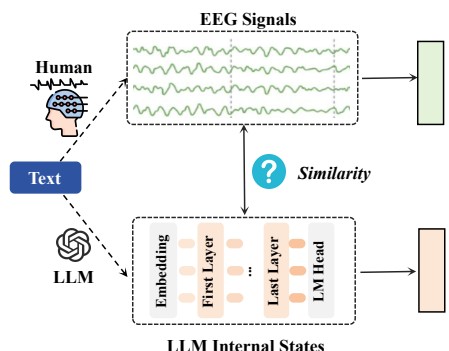

Figure 1: Comparison of human brain EEG signals and LLM internal states to explore similarities between human thought processes and model processing trajectories.

2024a; Mahner et al., 2025). While these comparisons suggest AI is becoming more human-like, they rely on behavioral data rather than neural evidence. Recently, research has shifted to exploring the alignment between AI mechanisms and human brain activity, particularly for LLMs. Neuroimaging techniques such as fMRI (Du et al., 2025), EEG (Xiao et al., 2025), and MEG (Wehbe et al.,

2014) have provided insights into the neural responses involved. Previous work has demonstrated alignment between LLMs and brain activity, primarily through linear mappings between neural responses and LLM representations (Zhou et al., 2024), such as activations, attention heads, and layer transformations (Caucheteux & King, 2022; Kumar et al., 2024b). These studies have examined various factors, including architectures and training conditions (Toneva & Wehbe, 2019; Mischler et al., 2024). LLM-brain alignment has also been used to investigate neural mechanisms, such as predictive processing and meaning composition, and to enhance both LLM performance and human-like language alignment (Rahimi et al., 2025; Moussa et al., 2024).

However, previous studies have largely focused on static correspondences or outcome-level similarities between LLM representations and neural responses, neglecting the temporal dynamics and processing trajectories that underpin human cognition. This raises an important question: *does this similarity stem solely from the convergent outputs of the models and the brain, or do these models emulate the underlying neural processing trajectory that govern human cognition*? This distinction is essential for understanding whether LLMs merely approximate brain activity or whether their internal computations reflect a deeper structural and functional resemblance to neural processes. As shown in Figure 1, our central motivation is to investigate this dynamic relationship by comparing the evolving EEG signals with the internal states of LLMs across layers, revealing the similarities and differences in their processing trajectories.

To investigate such similarities and differences between human cognitive processes and LLM computational trajectories, we conducted an experiment to compare EEG-extracted neural features with the text embeddings of 16 publicly available pretrained LLMs. Our analysis, based on English and Chinese texts, focused on two key aspects: **representational similarity** and **trajectory similarity**. To evaluate representational similarity, we used ridge regression (McDonald, 2009) and applied metrics like Pearson correlation, Representational Similarity Analysis (RSA) (Kriegeskorte et al., 2008; Kriegeskorte & Kievit, 2013; Diedrichsen & Kriegeskorte, 2017), and Centered Kernel Alignment (CKA) (Saha et al., 2022) to quantify the correspondences between EEG signals and LLM representations. To analyze trajectory similarity across layers and time, we introduced Latent Trajectory Comparison (LTC) to analyze the similarity between "neural trajectory" of brain responses and the corresponding "LLM latent trajectory" from various aspects, including magnitude, angular changes, uncertainty, and confidence evolution. Our findings show that middle-to-high layers of LLMs play a key role in semantic integration, aligning with the N400 component in EEG, a marker of semantic processing. This suggests LLMs capture brain-like processing for semantic understanding. However, while the brain processes language continuously, LLMs exhibit discrete bursts of activity. Cross-linguistic comparisons reveal that LLMs align better with EEG for English, while the alignment is weaker for Chinese, suggesting that LLMs trained mainly on English data may struggle with the subtleties of non-English languages. **In conclusion, LLMs partially emulate neural processing trajectories by capturing temporal dynamics and semantic integration patterns observed in EEG, showing that this similarity goes beyond convergent outputs, albeit more discretely and segmentally than the brain.** Our main contributions could be summarized as follows:

1. We systematically compare LLMs and human brain activity, evaluating 16 publicly available pretrained LLMs in English and Chinese texts. Using ridge regression to model LLM embeddings with EEG signals, we provide a large-scale, multilinguistic assessment of the similarity between LLM representations and neural activity in natural language processing.

2. Beyond static feature alignment, we analyse the temporal "neural trajectory" of brain responses and the corresponding "LLM latent trajectory" traced across hidden layers, incorporating measures of magnitude, angle, uncertainty, and confidence, providing insight into how dynamic neural processes relate to the evolving representations within LLMs.

3. Our analyses reveal that middle-to-high layers of LLMs generally serve as the core stage for hierarchical semantic integration. In contrast to the brain's continuous and iterative recalibration during reading, LLMs often process information in delayed, stage-end bursts, highlighting distinct strategies in real-time semantic processing.

## 2 RELATED WORK

**Neuroscientific Foundations of Language Comprehension.** The neuroscience of language comprehension investigates the spatiotemporal dynamics of brain-based linguistic processing, spanning

low-level perception to high-level semantic integration. Early work identified core "language network" regions, such as Broca's area (syntax/production) (Flinker et al., 2015) and Wernicke's area (semantics) (Ardila et al., 2016), but modern studies have refined this view to a distributed system. For example, fMRI research has shown that the left inferior frontal gyrus (LIFG), left middle temporal gyrus (LMTG), and angular gyrus (AG) collectively resolve syntactic ambiguities and integrate word meanings into coherent propositions (Noonan et al., 2013). With millisecond-level temporal resolution, EEG has further illuminated the timing of language processing via event-related potentials (ERPs) (Van Berkum et al., 2005). The N400 component (400 ms post-stimulus) responds to semantic anomalies, reflecting efforts to integrate unexpected words (Fogelson et al., 2004), and the P600 index syntactic reanalysis (Tanner et al., 2017). These ERPs act as neurophysiological markers for linguistic representation building, revealing intermediate processing steps overlooked by behavioral measures. Collectively, these findings establish that language comprehension is a dynamic, incremental process shaped by both bottom-up sensory input and top-down contextual expectations.

**Brain Similarity of Language Models.** Numerous studies have shown that deep neural network representations can be linearly mapped to neural responses (Toneva & Wehbe, 2019; Schrimpf et al., 2021; Anderson et al., 2021), suggesting that both human brains and language models are involved in predicting the next word (Schrimpf et al., 2021). Brain activation correlates with language models, peaking around 400 ms after word onset (Goldstein et al., 2022). Further work has explored aspects like autoregressive models (Goldstein et al., 2022; Caucheteux et al., 2023), model size, and linguistic generalizability (Caucheteux & King, 2022; Antonello & Huth, 2024), providing insights into the brain-like nature of language processing in LLMs. As models trained on massive text corpora, LLMs demonstrate emergent abilities in semantic parsing, context integration, and hierarchical processing (Li et al., 2024). Notably, embeddings from later LLM layers have been shown to correlate with fMRI and MEG responses during language comprehension, indicating partial alignment between computational and neural semantic representations (Zhou et al., 2024; Mischler et al., 2024; Nakagi et al., 2024; Rahimi et al., 2025; Lei et al., 2025; Du et al., 2025). For example, Ren et al. (2024) and Du et al. (2025) employed representational similarity analysis (RSA) to compare text embeddings with fMRI signals, constructing representational dissimilarity matrices (RDMs) via metrics such as Pearson correlation. Other studies (Zhou et al., 2024) aligned layerwise activations of language models with averaged MEG activity maps via ridge regression (McDonald, 2009). Additionally, Tuckute et al. (2024) trained encoding models on fMRI data from participants exposed to diverse sentences, optimizing GPT-2 XL embeddings to enhance neural alignment. Unlike most existing studies that rely on static analysis, we differentiate our approach by quantifying dynamic alignment, offering a deeper understanding of the evolving EEG and LLM patterns and highlighting both shared and unique aspects of their interactions.

## 3 METHODOLOGY

To investigate the similarity between LLM representations and human neural activity during language comprehension, as summarized in Figure 3, we investigate two types of similarity: (1) for **representation similarity**, we predict EEG features from LLM embeddings using ridge regression and assess alignment through Pearson correlation, RSA, spatiotemporal alignment, and functional connectivity. (2) for **trajectory similarity**, we apply latent trajectory comparison (LTC) to examine "neural trajectory" and "LLM latent trajectory" through various measures, including magnitude variations, angular shifts, uncertainty fluctuations, and confidence evolution.

### 3.1 REPRESENTATION SIMILARITY

To assess the alignment between LLM representations and human neural activity, we first assess representation similarity. Specifically, we process the text by segmenting it into sentences and feeding them into 16 pretrained LLMs. To quantify how semantic representations from different layers of LLMs relate to EEG activity, we employ ridge regression in a layerwise encoding framework. Let $M \in \mathbb{R}^{N \times d}$ denote the EEG responses and let $L \in \mathbb{R}^{N \times L \times D}$ denote the LLM embeddings, where $N$ is the number of samples, $L$ is the number of layers, and $D$ is the embedding dimensionality. For each layer $l$ and fold $k$ in $K$-fold cross-validation, the ridge regression weights are estimated as:

$$\hat{W}^{(l,k)} = \left(L_{\text{train}}^{(l,k)^\top} L_{\text{train}}^{(l,k)} + \alpha I\right)^{-1} L_{\text{train}}^{(l,k)^\top} M_{\text{train}}^{(k)}, \tag{1}$$

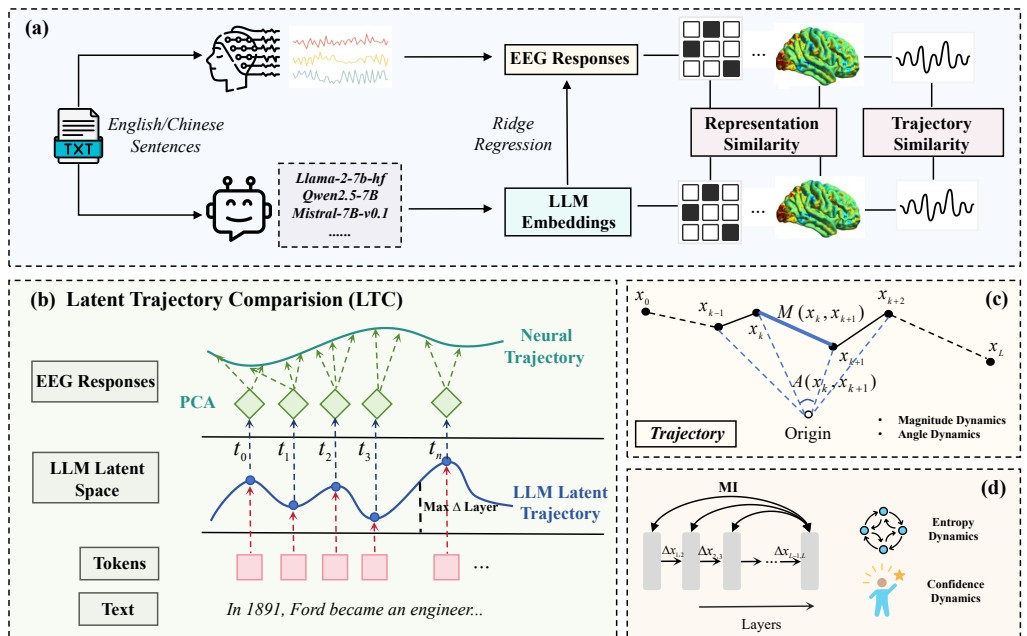

Figure 2: Overview of the proposed methodology for investigating brain-LLM language processing similarities. (a) Framework for measuring Representation similarity: Pearson correlation (ridge regression), spatiotemporal (ST) alignment, and latent trajectory comparison (LTC). (b) LTC: Trajectories across layers and time are compared. (c) Magnitude and angular dynamics: Analysing intensity and directionality. (d) Uncertainty and confidence dynamics.

where $\alpha$ is the regularization parameter selected via nested cross-validation. The predicted EEG responses for the test set are as follows:

$$\hat{M}_{i,\text{test}} = L_{i,\text{test}}\hat{W}^{(l,k)} + \hat{b}_i. \tag{2}$$

To quantify the representational similarity between predicted EEG $\hat{M}$ and ground-truth EEG $M$, we employ RSA by computing RDMs (Kriegeskorte et al., 2008; Kriegeskorte & Kievit, 2013; Diedrichsen & Kriegeskorte, 2017) for both and measuring similarity as the Spearman correlation between the upper triangular elements of $\text{RDM}_M$ and $\text{RDM}_{\hat{M}}$, yielding an RSA score reflecting how well the predicted responses preserve the representational structure of true EEG. To capture global subspace alignment, we compute CKA (Saha et al., 2022):

$$\text{CKA}(\hat{M}, M) = \frac{\|\hat{M}^\top M\|_F^2}{\|\hat{M}^\top \hat{M}\|_F \cdot \|M^\top M\|_F}, \tag{3}$$

where $\|\cdot\|_F$ is the Frobenius norm and where $\hat{M}, M$ is mean-centered.

As a sanity check, we evaluate whether the predictive model could accurately capture both the spatial and temporal dynamics of language processing. We assess spatiotemporal alignment between EEG signals and LLM predictions by computing time-resolved, channelwise correlations to generate topographic maps. Functional connectivity (Fingelkurts et al., 2005) is quantified via Pearson correlations across channels within sliding time windows for both EEG and LLM-predicted responses. This captures the temporal evolution of neural activity and enables network-level comparisons.

## 3.2 LATENT TRAJECTORY COMPARISON

Building on representation similarity, we further explore trajectory similarity to capture the dynamic evolution of information processing in both brains and LLMs. We compare the neural trajectory and

LLM latent trajectory, tracking semantic evolution through measures including magnitude and angle changes, uncertainty, mutual information, skewness, kurtosis, Lyapunov exponent, and dynamic alignment, providing a holistic view of processing in both systems.

**Trajectory Formalization.** For both EEG and LLM, the "trajectory" is defined as a sequence of transformations across temporal steps or layers. The unified trajectory can be expressed as:

$$\boldsymbol{H} = \underbrace{\boldsymbol{h}_0}_{\text{initial state}} \rightarrow \underbrace{\boldsymbol{h}_1 \rightarrow \cdots \rightarrow \boldsymbol{h}_l \rightarrow \cdots \rightarrow \boldsymbol{h}_{L-1}}_{\text{intermediate states}} \rightarrow \underbrace{\boldsymbol{h}_L}_{\text{final state}}, \tag{4}$$

where for EEG, each $\boldsymbol{h}_l$ represents the neural state at temporal window $l$, and for LLM, each $\boldsymbol{h}_l$ denotes the hidden state at layer $l$.

**Magnitude and Angle Dynamics.** We compare the geometric features of the EEG and LLM trajectory by examining the magnitude and angle changes between adjacent states in the embedding trajectory. Both the magnitude change $M(\boldsymbol{h}_l, \boldsymbol{h}_{l+1})$ and the angle change $A(\boldsymbol{h}_l, \boldsymbol{h}_{l+1})$ are:

$$M(\boldsymbol{h}_l, \boldsymbol{h}_{l+1}) = \|\boldsymbol{h}_{l+1} - \boldsymbol{h}_l\|_2, \quad A(\boldsymbol{h}_l, \boldsymbol{h}_{l+1}) = \arccos\left(\frac{\boldsymbol{h}_{l+1}^\top \boldsymbol{h}_l}{\|\boldsymbol{h}_{l+1}\|_2 \|\boldsymbol{h}_l\|_2}\right). \tag{5}$$

where $M(\boldsymbol{h}_l, \boldsymbol{h}_{l+1})$ quantifies the distance between consecutive states, and $A(\boldsymbol{h}_l, \boldsymbol{h}_{l+1})$ measures the angular change, which indicates the directional shift in the trajectory.

To normalize the absolute changes across different trajectories, we define the average magnitude and angle over the entire trajectory as follows:

$$\text{Mag}(\boldsymbol{H}) = \frac{1}{L} \sum_{l=0}^{L-1} \frac{M(\boldsymbol{h}_l, \boldsymbol{h}_{l+1})}{\mathcal{Z}_{\text{Mag}}}, \quad \text{Ang}(\boldsymbol{H}) = \frac{1}{L} \sum_{l=0}^{L-1} \frac{A(\boldsymbol{h}_l, \boldsymbol{h}_{l+1})}{\mathcal{Z}_{\text{Ang}}}, \tag{6}$$

where $\mathcal{Z}_{\text{Mag}}$ and $\mathcal{Z}_{\text{Ang}}$ are scaling factors used to normalize the absolute magnitude and angle changes relative to the overall trajectory.

**Uncertainty and Confidence Dynamics.** The dynamics of uncertainty and confidence reveal how a system accumulates and processes information over time. We focus on matrix-based entropy (Yu et al., 2021; Skean et al., 2025), a comprehensive metric that quantifies uncertainty while considering both compression and variability in the system's representations (see Appendix A.1 for details). Let $Z \in \mathbb{R}^{N \times D}$ be the matrix of hidden states at time step $k$. We define the Gram matrix $K = ZZ^\top$, which captures pairwise relationships between data points. The matrix-based entropy $S_\alpha(Z)$ for order $\alpha > 0$ is as follows:

$$S_\alpha(Z) = \frac{1}{1-\alpha} \log\left(\sum_{i=1}^{r} \left(\frac{\lambda_i(K)}{\text{tr}(K)}\right)^\alpha\right) \tag{7}$$

where $r = \text{rank}(K) \leq \min(N, D)$, $\lambda_i(K)$ are the eigenvalues of $K$, and $\text{tr}(K)$ is the trace. We typically use $\alpha = 1$ for simplicity, as it simplifies the entropy measure to von Neumann entropy.

Confidence can be interpreted as the inverse of uncertainty, providing a complementary view of system dynamics. For each stage:

$$C^{(X)}(k) = \frac{1}{S_\alpha(Z) + \epsilon} \bigg/ \max_{k'} \frac{1}{S_\alpha(Z) + \epsilon} \tag{8}$$

This normalizes confidence to a 0–1 scale, with $\epsilon = 10^{-8}$, ensuring numerical stability.

**Mutual Information.** Mutual information (MI) (Kraskov et al., 2004) measures the shared information between two variables, reflecting their dependence. In both EEG signals and LLMs, MI captures the relationship between intermediate layers and the final output, revealing how information propagates. For both EEG signals and LLMs, the mutual information between an intermediate layer $\boldsymbol{h}_i$ and the final output $\boldsymbol{h}_L$ is given by:

$$I(\boldsymbol{h}_i, \boldsymbol{h}_L) = \sum_{x \in \boldsymbol{h}_i} \sum_{y \in \boldsymbol{h}_L} p(x, y) \log\left(\frac{p(x, y)}{p(x)p(y)}\right) \tag{9}$$

where $\boldsymbol{h}_i$ refers to the intermediate layer embedding and $\boldsymbol{h}_L$ represents the final output layer.

**Skewness, Kurtosis, and Lyapunov Exponent.** Skewness and kurtosis (Groeneveld & Meeden, 1984) quantify EEG and LLM feature asymmetry and peakness, while the Lyapunov exponent (Young, 2013) measures sensitivity to initial conditions, with positive values indicating chaos.

**Dynamic Representational Alignment (DRA).** We propose a metric DRA (Appendix A.3) on the Hilbert space $\boldsymbol{H}$ (Young, 1988), where EEG representations $\boldsymbol{E}(t) \in \boldsymbol{H}_{\text{EEG}}^d$ and LLM hidden states $\boldsymbol{L}(t) \in \boldsymbol{H}_{\text{LLM}}^k$ have bounded norms. DRA incorporates Gaussian distribution divergence to penalize shifts and applies a probabilistic weight to emphasize important time steps. The formulation is:

$$\text{DRA} = \frac{1}{Z_T} \sum_{t=1}^{T} \omega(t) \cdot \cos(\boldsymbol{E}(t), \boldsymbol{L}(t)) \cdot \frac{\langle \Delta \boldsymbol{E}(t), \Delta \boldsymbol{L}(t) \rangle_{\boldsymbol{H}}}{|\Delta \boldsymbol{E}(t)|_{\boldsymbol{H}} |\Delta \boldsymbol{L}(t)|_{\boldsymbol{H}} + \epsilon} \cdot e^{-\alpha \cdot \text{KL}(P_t \| Q_t)} \quad (10)$$

where $Z_T$ is an $\ell_2$-normalization factor to keep DRA in $[0,1]$; $\omega(t) \propto \text{Gamma}(t; \beta, 1)$ $(\beta > 0)$ weights time-step importance; $P_t = \mathcal{N}(\mu_{\boldsymbol{E}(t)}, \Sigma_{\boldsymbol{E}(t)})$ and $Q_t = \mathcal{N}(\mu_{\boldsymbol{L}(t)}, \Sigma_{\boldsymbol{L}(t)})$ are EEG or LLM Gaussian representations with $\text{KL}(\cdot \| \cdot)$ the Kullback-Leibler divergence; $\epsilon = 10^{-8}$ ensures numerical stability; $\alpha \in (0, 5]$ controls the divergence penalty.

# 4 EXPERIMENTS AND RESULTS

## 4.1 DATA PREPARATION AND PREPROCESSING

Many studies linking brain activity and LLMs have used fMRI (Karamolegkou et al., 2023; Oota & Bapi, 2024), but its low temporal resolution limits tracking word-level processing. To overcome this, we use EEG for the first time, which capture millisecond-level neural dynamics during language comprehension, across two datasets: (1) **ZuCo Dataset** (Hollenstein et al., 2018): English EEGs and eye-tracking data from 12 participants reading 1,050 sentences (movie reviews and Wikipedia) under normal reading (NR). The data were recorded with a 128-channel Geodesic Hydrocel system at 500 Hz and preprocessed with artifact rejection, interpolation, and rereferencing. (2) **ChineseEEG Dataset** (Mou et al., 2024): Chinese text reading EEGs from 10 participants (The Little Prince and Garnett Dream, 115,233 characters) using a 128-channel system at 1 kHz, preprocessed with segmentation, downsampling, filtering, ICA denoising, and referencing. In both datasets, each sentence corresponds to a distinct segment of EEG signals, establishing a direct one-to-one mapping between linguistic stimuli and neural responses.

To provide a comprehensive evaluation across diverse architectures, we employ a set of sixteen state-of-the-art, publicly available LLMs from the HuggingFace[1] repositories. These models span multiple families (LLaMA (Touvron et al., 2023), Qwen (Hui et al., 2024), Mistral (Siino, 2024), Gemma (Team et al., 2024), Falcon (Almazrouei et al., 2023), Yi (Young et al., 2024), DeepSeek (Bi et al., 2024)) and cover both *base* and *instruction-tuned* variants (see Appendix A.4).

## 4.2 MODEL CORRELATION PERFORMANCE

For sentence-level EEG analysis, we aggregated neural responses by averaging across all time steps and channels corresponding to each sentence, producing a unified representation per sentence and participant. Table 1 summarizes the similarity results for 16 LLMs, evaluated using mean squared error (MSE), Pearson correlation ($r$), representational similarity analysis (RSA), and centered kernel alignment (CKA). On the ZuCo dataset, gemma-7b-it achieved the highest Pearson correlation of 0.5103, while Meta-Llama-3-8B-Instruct attained the best CKA score of 0.4350. Instruction-tuned variants consistently outperformed their base models, indicating that instruction tuning improves representational alignment with neural responses. In contrast, Pearson correlation on the ChineseEEG dataset was generally lower. Yi-1.5-9B had the lowest MSE of 1.2072, while Llama-2-7b-hf scored highest in correlation and CKA. Mistral-7B-Instruct-v0.1 achieved the best RSA score. Unlike the English dataset, base models often outperformed instruction-tuned variants on Chinese, likely due to limited high-quality Chinese instruction data and the predominance of English optimization in instruction tuning, leading to mismatches with Chinese linguistic and cultural nuances.

As shown in Figure 3 (a), representational dissimilarity matrices computed from EEG and ridge-regressed LLM predictions via Euclidean distances exhibit similar spatial patterns. The correlation

---

[1]https://huggingface.co/

Table 1: Sentence-level alignment results between LLM representations and brain signals for both English and Chinese datasets. Best results per column are in **bold**.

| Model | ZuCo Dataset | | | | ChineseEEG Dataset | | | |
|---|---|---|---|---|---|---|---|---|
| | MSE ↓ | $r$ ↑ | RSA ↑ | CKA ↑ | MSE ↓ | $r$ ↑ | RSA ↑ | CKA ↑ |
| Llama-2-7b-hf | 0.8370 | 0.4809 | 0.3987 | 0.3967 | 1.1821 | **0.1675** | 0.1354 | **0.3936** |
| Llama-2-7b-chat-hf | 0.8340 | 0.4951 | **0.4360** | 0.3931 | 1.2163 | 0.1320 | 0.1298 | 0.3762 |
| Meta-Llama-3-8B | 0.8257 | 0.4980 | 0.4044 | 0.4125 | 1.2157 | 0.1475 | 0.1381 | 0.3697 |
| Meta-Llama-3-8B-Instruct | **0.8128** | 0.5026 | 0.4220 | **0.4350** | 1.2194 | 0.1349 | 0.1293 | 0.3429 |
| Qwen2.5-7B | 0.9834 | 0.3828 | 0.2064 | 0.2841 | 1.2639 | 0.0702 | 0.1086 | 0.2794 |
| Qwen2.5-7B-Instruct | 0.9806 | 0.3832 | 0.2068 | 0.2778 | 1.2564 | 0.0789 | 0.1120 | 0.2946 |
| Mistral-7B-v0.1 | 0.8117 | 0.4681 | 0.3477 | 0.4169 | 1.2218 | 0.1210 | 0.1172 | 0.3737 |
| Mistral-7B-Instruct-v0.1 | 0.8268 | 0.4714 | 0.3852 | 0.4127 | 1.2171 | 0.1338 | **0.1410** | 0.3887 |
| gemma-7b | 0.8678 | 0.4841 | 0.4160 | 0.3990 | 1.2331 | 0.1552 | 0.1379 | 0.3127 |
| gemma-7b-it | 0.8140 | **0.5103** | 0.3824 | 0.3852 | 1.2444 | 0.1308 | 0.1084 | 0.2815 |
| Falcon3-7B-Base | 0.8855 | 0.4416 | 0.3481 | 0.3689 | 1.2130 | 0.1098 | 0.1116 | 0.3553 |
| Falcon3-7B-Instruct | 0.8842 | 0.4396 | 0.3368 | 0.3685 | 1.2298 | 0.1493 | 0.1352 | 0.3435 |
| Yi-1.5-9B | 0.8679 | 0.4302 | 0.2909 | 0.3481 | **1.2072** | 0.1218 | 0.1105 | 0.3754 |
| Yi-1.5-9B-Chat | 0.8937 | 0.4508 | 0.3097 | 0.2969 | 1.2663 | 0.0489 | 0.0697 | 0.2536 |
| deepseek-llm-7b-base | 0.8261 | 0.4886 | 0.3822 | 0.4021 | 1.2510 | 0.0751 | 0.0888 | 0.2436 |
| DeepSeek-R1-Distill-Qwen-7B | 0.9919 | 0.3554 | 0.2593 | 0.3104 | 1.2375 | 0.1029 | 0.0690 | 0.2386 |

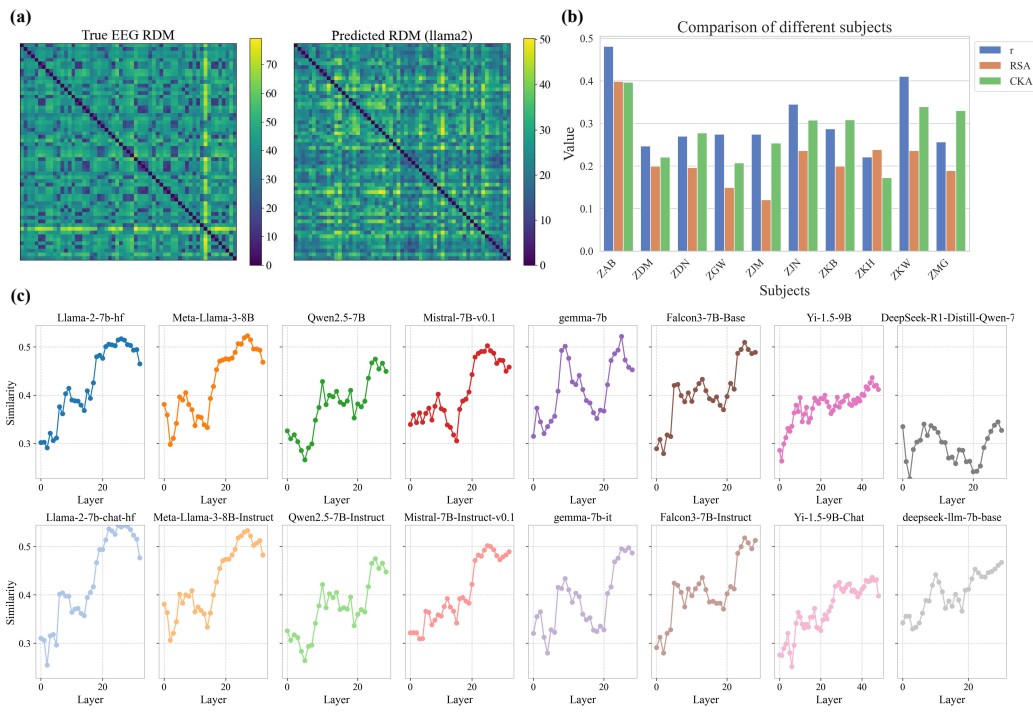

Figure 3: Similarity analysis. (a) Visualization of EEG-LLM similarity via RDMs. (b) Comparison across different subjects. (c) Trend of similarity between LLM layers and EEG responses.

of the upper-triangular elements revealed a significant positive relationship ($R = 0.4066$, $p < 0.05$), indicating that LLMs partially capture human representational structures. Figure 3 (b) presents the results of similarity analysis for different subjects. subject ZAB has the highest values for all the metrics, whereas ZIM has a relatively low RSA. Generally, Pearson correlation tends to be greater than RSA and CKA, suggesting that the Pearson correlation captures stronger neural-model associations in this study. Figure 3 (c) presents the similarity curves between different layers of the LLMs and EEG signals, the layer similarity curves of all the models exhibit nonmonotonic fluctuations, with peaks typically occurring in the middle-to-high layers (10–30). These findings

suggest that these layers play a key role in integrating in-depth features during the hierarchical semantic processing of LLMs.

To quantify how much the observed alignment exceeds chance levels, we conducted baseline evaluations under two randomization conditions: (1) permuting EEG-text pairs to disrupt natural correspondences and (2) shuffling the feature dimensions of LLM embeddings. As shown in the Table 4.2, the performance under the true condition (using Llama-2-7b-hf) substantially exceeds both random baselines across all the metrics, confirming that the observed alignment is statistically significant and not attributable to chance.

Table 2: Statistical significance of observed alignment (Llama-2-7b-hf) against random baselines.

| Condition | MSE | Pearson correlation | RSA | CKA |
|---|---|---|---|---|
| True condition (Ours) | 0.8370 | 0.4809 | 0.3987 | 0.3967 |
| Random: Permuted Pairs | 1.4257 | -0.0436 | 0.0839 | 0.1165 |
| Random: Shuffled Embeddings | 1.8833 | 0.0066 | 0.0753 | 0.1129 |

### 4.3 SPATIOTEMPORAL PATTERNS OF PREDICTIONS

The EEG-LLM correlation topomaps in Figure 4 (a) reveal dynamic spatial patterns of similarity across time. In the early stage (0–200 ms), EEG shows positive correlations for sensory processing, with negative correlations around 100 ms indicating categorization and filtering. In the mid-stage (200-400 ms), significant positive correlations appear, particularly around 300 ms, corresponding to semantic integration and syntactic analysis. It suggests that LLMs simulate the brain's semantic network and syntactic processing. In the later stage (400–500 ms), central–anterior effects at 400 ms align with the N400 component, reflecting semantic integration during language comprehension. EEG topographies show involvement of key language areas, such as Broca's and Wernicke's areas, aligning with LLM's attention mechanisms. Hemispheric asymmetry, especially right-side correlations at 300 ms, mirrors the lateralization of language processing, indicating a correspondence between LLMs and brain hemispheric specialization (Van Berkum et al., 2005; Tanner et al., 2017).

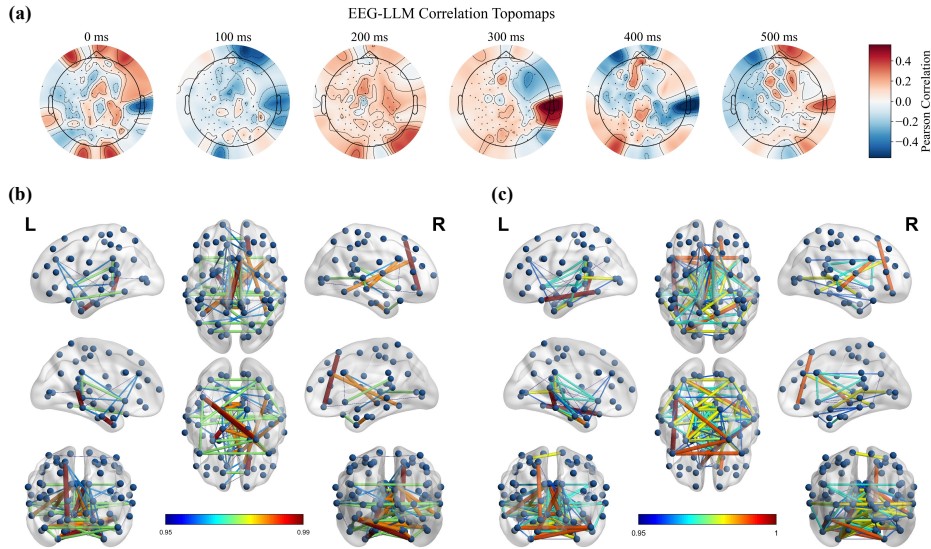

Figure 4: Topographic maps and connectivity analysis. (a) Topographic maps of EEG–LLM correlations. (b) EEG functional connectivity patterns. (c) Functional connectivity predicted by LLM.

EEG functional connectivity shows sparse but strong links among core regions with weak global coupling, reflecting "functional differentiation with efficient coordination." In contrast, LLM-predicted connectivity is densely distributed, suggesting "global generalization with diminished

regional specificity" and limited fidelity to biological networks. Both modalities highlight strong central and temporal language-related connectivity, indicating that LLMs capture the core collaborative network for language. However, weak frontal–occipital links in EEG (Figure 4 (b)) are overestimated in LLMs (Figure 4 (c)), and temporal–limbic connections are underrepresented, underscoring insufficient modelling of cross-functional coordination and limbic contributions.

## 4.4 LATENT TRAJECTORY COMPARISON

**Magnitude and Angle Patterns.** As shown in Figure 4.4 (a) and (b), the features reveal distinct temporal dynamics between EEGs and LLMs. In terms of magnitude, EEGs exhibit continuous fluctuations with early peaks at steps 5 and 17, reflecting rapid, distributed, and iterative neural processing, whereas LLMs remain largely stable before a sharp surge at step 31, resembling a "silent analysis followed by late integration." Angle patterns show a similar divergence: EEGs display irregular peaks at steps 5, 12, and 25, which is consistent with ongoing neural reorientation, whereas LLMs rise gradually and spike only at step 31, suggesting sequential and hierarchical adjustment. Together, these results highlight a contrast between the brain's real-time semantic recalibration and the model's delayed, stage–end consolidation (see Appendix A.6 for ChineseEEG datasets).

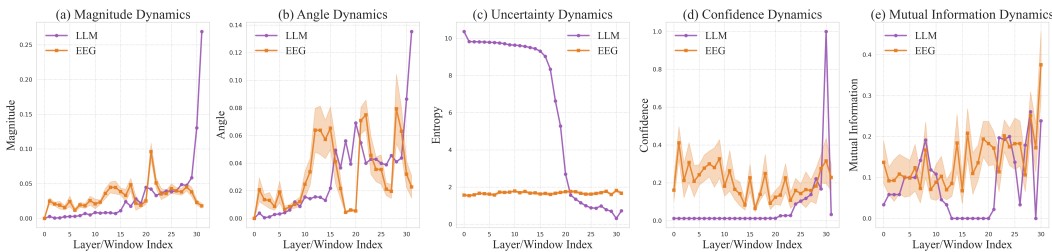

Figure 5: Temporal and dynamic comparisons between EEGs and LLMs. (a) Magnitude dynamics, (b) Angle dynamics, (c) Uncertainty dynamics, (d) Confidence dynamics, (e) MI dynamics. The shaded area represents the standard error of multiple subjects, which is used to measure the range of data fluctuations.

**Uncertainty and Confidence Dynamics.** As shown in Figure 4.4 (c) and (d), LLMs start with high entropy, which rapidly decreases, whereas confidence gradually increases and peaks around Layer 30, reflecting a delayed, stage-like consolidation of uncertainty resolution. In contrast, EEGs maintain relatively stable entropy fluctuations alongside frequent confidence peaks and troughs, which is consistent with continuous real-time adjustment. The vertical dashed lines highlight critical transition points, underscoring the divergence between the brain's dynamic recalibration and the model's late integration strategy.

**MI Dynamics.** The MI dynamics shown in Figure 4.4 (e) reveal a clear divergence in information coupling: EEG show sharp, high-amplitude peaks, whereas LLM data display a gradual, low-amplitude rise, indicating distinct

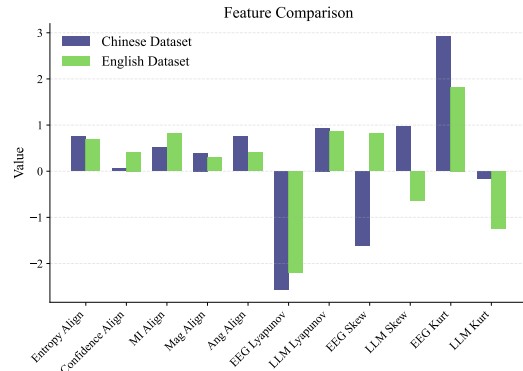

Figure 6: Alignment comparison across features between the English and Chinese datasets.

temporal modes of information integration during language processing.

**Alignment.** This section compares the alignment between the neural trajectory and the LLM latent trajectory for English and Chinese, using metrics including entropy, magnitude, skewness, kurtosis, and others to assess EEG–LLM correspondence. As shown in Figure 6, the alignment shows clear language-dependent differences. Entropy alignment is higher in Chinese, indicating stronger

structural similarity. MI alignment is higher in English, reflecting tighter information coupling. Magnitude and angular alignment are elevated in Chinese, suggesting stronger directional and amplitude consistency. Lyapunov exponents indicate slightly greater EEG instability in Chinese data, while LLM trajectories remain stable. Distributional properties further differ: Chinese EEG signals are negatively skewed with higher kurtosis, whereas LLM features show positive skew and lighter tails.

### 4.5 DISCUSSION

We investigate how LLMs simulate human neural trajectories from three complementary perspectives. (1) **Correlations and Spatiotemporal Patterns.** Activations in intermediate layers of LLMs exhibit higher correlations with EEG signals than those in final layers, consistent with Mischler et al. (2024). Our use of EEG complements prior fMRI (Lei et al., 2025) and MEG (Zhou et al., 2024) studies, offering millisecond-level resolution for tracking language processing. On the English ZuCo dataset, instruction-tuned LLMs outperform base models in both representational similarity and sentence comprehension, supporting (Oota et al., 2025). In contrast, for Chinese EEG data, base models often show better alignment, likely reflecting limited high-quality Chinese instruction-tuning data and highlighting language-specific constraints. While previous studies have investigated primarily the relationship between model scale and brain similarity (Bonnasse-Gahot & Pallier, 2024), our spatiotemporal analyses show that LLMs capture key neural landmarks such as the N400 component around 400 ms and central–temporal connectivity patterns. However, they overestimate frontal–occipital interactions and underrepresent temporal–limbic connections, indicating gaps in cross-network coordination and affective contributions. Li et al. (2023) have achieved word-level temporal modeling through sophisticated smoothing techniques, enabling a more fine-grained alignment between model representations and neural response measurements using fMRI. In contrast, our work focuses on sentence-level EEG analysis, establishing direct correspondence between complete sentences and their corresponding neural signatures. (2) **Latent Trajectory Metrics.** Analyses of magnitude and angle reveal dynamic differences. EEG responses exhibit continuous, iterative fluctuations with early peaks, whereas LLMs follow a staged pattern of silent analysis followed by late integration. Magnitude captures the intensity of state changes, analogous to neural activation fluctuations, and angle reflects directional transitions between cognitive stages, such as syntactic and semantic integration. Additional metrics, including uncertainty, confidence, and mutual information, indicate that the human brain updates continuously while LLMs respond in discrete, stepwise stages. Together, these results show that LLMs replicate the core temporal and stepwise dynamics of neural processing, although in a more discrete and segmented manner. (3) **Cross-linguistic Comparisons.** LLMs simulate neural trajectories more accurately in English than in Chinese. English, with its root-word and syntactic structures, aligns better with token-based LLM processing, whereas Chinese, with its logographic and context-dependent features, presents greater challenges (Chen et al., 2025). It should be noted that these comparative conclusions are influenced by multiple factors including different experimental datasets, text materials, and participant groups across the two languages. Although overall alignment metrics are comparable across languages, dynamical and statistical properties such as Lyapunov exponents, skewness, and kurtosis differ, reflecting language-specific structural influences on both neural and model dynamics.

## 5 CONCLUSION

In this work, we present a cross-linguistic assessment of the similarity between human brain activity and LLMs. By comparing 16 publicly available pretrained LLMs with human EEG responses during natural language processing tasks in both English and Chinese, we evaluated their similarity from the perspectives of representational similarity and trajectory similarity. We used ridge regression to quantify the alignment between LLM embeddings and EEG signals, and further analyzed the trajectory evolution of information processing. Our findings show that middle-to-high layers of LLMs are crucial for semantic integration, and while the brain continuously adjusts during reading, LLMs often process information in discrete, stage-end bursts. This study offers valuable insights into both the shared and distinct computational strategies of the brain and LLMs, contributing to the development of more human-like models.

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

## A APPENDIX

### A.1 MATRIX-BASED ENTROPY

A key advantage of matrix-based entropy is that it provides a unified perspective on multiple aspects of representation quality in LLM embeddings.

**1. Compression and Information Content.** If only a few eigenvalues are large, $K$ is approximately low-rank, indicating that the model has condensed input variation into a smaller subspace (Skean et al., 2025). Conversely, a more uniform spectrum corresponds to higher entropy and more diverse features.

**2. Geometric Smoothness.** The trajectory of embeddings across tokens can exhibit curvature in the representation space. Sharp local turns correspond to skewed eigenvalue distributions (Hosseini & Fedorenko, 2023), whereas smooth trajectories yield more evenly distributed eigenvalues. This captures not only token-to-token transitions but also longer-range structural patterns across segments or entire prompts.

**3. Invariance under Augmentations.** Representational stability under augmentations can be assessed via the clustering structure in $K$. Strong invariance manifests as stable clusters in $ZZ^\top$, reflecting the retention of meaningful global structure while potentially discarding irrelevant local variations (Skean et al., 2025).

### A.2 THEORETICAL VALIDITY OF TRAJECTORY FORMALIZATION AND MAGNITUDE–ANGLE DYNAMICS

Chain formalization is theoretically justified by the stagewise evolution paradigm shared across systems. Both EEG and LLM information processing follow an "initial input $\rightarrow$ intermediate transformations $\rightarrow$ final output" logic. Trajectory capture this via discrete state sequences. For an EEG, the temporal evolution can be represented as

$$h_0^{\text{EEG}} \rightarrow h_1^{\text{EEG}}, \ldots, h_{L-1}^{\text{EEG}} \rightarrow h_L^{\text{EEG}}, \tag{11}$$

where $h_0^{\text{EEG}}$ encodes sensory input (e.g., initial visual cortex activation), $h_1^{\text{EEG}}, \ldots, h_{L-1}^{\text{EEG}}$ represent feature integration (e.g., associative cortical fusion), and $h_L^{\text{EEG}}$ denotes cognitive output (e.g., decision-related activation). State transitions satisfy the continuity assumption of neural dynamics: $h_{l+1}^{\text{EEG}}$ depends only on $h_l^{\text{EEG}}$, which is consistent with ERP temporal locking (Pradhan & Dutt, 1993). For LLMs, hierarchical evolution is captured as

$$h_0^{\text{LLM}} \rightarrow h_1^{\text{LLM}}, \ldots, h_k^{\text{LLM}} \rightarrow h_{k+1}^{\text{LLM}}, \ldots, h_{L-1}^{\text{LLM}} \rightarrow h_L^{\text{LLM}}, \tag{12}$$

where $h_0^{\text{LLM}}$ is the input embedding, $h_1^{\text{LLM}}, \ldots, h_k^{\text{LLM}}$ encode low-level syntactic features, $h_{k+1}^{\text{LLM}}, \ldots, h_{L-1}^{\text{LLM}}$ abstract high-level semantics, and $h_L^{\text{LLM}}$ generates output. Layerwise transitions

follow the locality assumption of attention, which is consistent with empirical findings (Wang et al., 2024).

Mathematically, the state sequences are both measurable and complete. Denote the state space as $\mathbb{R}^D$ ($D$-dimensional embeddings) and the trajectory $\boldsymbol{H} = \{\boldsymbol{h}_0, \boldsymbol{h}_1, \ldots, \boldsymbol{h}_L\}$. Using the Euclidean distance $d(\boldsymbol{a}, \boldsymbol{b}) = \|\boldsymbol{a} - \boldsymbol{b}\|_2$:

1. Nonnegativity: $d(\boldsymbol{h}_l, \boldsymbol{h}_m) \geq 0$, with equality iff $\boldsymbol{h}_l = \boldsymbol{h}_m$.

2. Symmetry: $d(\boldsymbol{h}_l, \boldsymbol{h}_m) = d(\boldsymbol{h}_m, \boldsymbol{h}_l)$.

3. Triangle inequality: $d(\boldsymbol{h}_l, \boldsymbol{h}_n) \leq d(\boldsymbol{h}_l, \boldsymbol{h}_m) + d(\boldsymbol{h}_m, \boldsymbol{h}_n)$ for $l < m < n$.

These follow directly from the properties of Euclidean distance (Krislock & Wolkowicz, 2012). As the EEG time interval $\Delta t \to 0$, the trajectory limit $\boldsymbol{H}^{\text{EEG}}$ approaches a continuous function $\boldsymbol{h}^{\text{EEG}}(t) : [0, T_{\text{total}}] \to \mathbb{R}^D$, which is uniformly continuous due to the limited EEG bandwidth. Similarly, as the LLM depth $L \to \infty$, $\boldsymbol{H}^{\text{LLM}}$ converges to a continuous mapping $\boldsymbol{h}^{\text{LLM}}(x) : [0, 1] \to \mathbb{R}^D$, guaranteeing completeness.

**Magnitude Changes** quantify the "strength of information update." For EEG, $M$ correlates with the event-related desynchronization (ERD) amplitude (Yordanova et al., 2001); a larger $M$ indicates stronger neural updates (e.g., P300 component). For LLMs, $M$ measures interlayer semantic gain: low layers exhibit larger $M$ (rapid syntactic generation), and high layers have smaller $M$ (semantic stabilization) Momo et al. (2024). $M$ satisfies monotonicity with respect to $\|\Delta \boldsymbol{h}\|_2$ and additivity:

$$M(\boldsymbol{h}_l, \boldsymbol{h}_{l+2}) \leq M(\boldsymbol{h}_l, \boldsymbol{h}_{l+1}) + M(\boldsymbol{h}_{l+1}, \boldsymbol{h}_{l+2}), \tag{13}$$

with equality if successive changes are collinear.

**Angle Changes** measure the directional deviation. For an EEG, a small $A$ indicates task-aligned evolution; a large $A$ indicates perturbation. For LLMs, a small $A$ indicates coherent semantic generation; a large $A$ indicates divergence. $A$ satisfies

$$A \in [0, \pi], \quad A(k_1 \boldsymbol{h}_l, k_2 \boldsymbol{h}_{l+1}) = A(\boldsymbol{h}_l, \boldsymbol{h}_{l+1}) \, \forall k_1, k_2 > 0, \tag{14}$$

showing boundedness and scale invariance.

A.3 ALIGNMENT METRIC

To validate the effectiveness and reliability of the proposed DRA metric in quantifying EEG-LLM trajectory alignment, we prove three key theoretical properties: monotonicity (consistency with similarity trends), robustness (insensitivity to bounded noise), and normalization (range constraint to $[0, 1]$).

1. Proof of Monotonicity

**Proposition:** If for all time steps $t \in \{1, 2, \ldots, T\}$, the trajectory coherence term satisfies

$$\frac{\langle \Delta \boldsymbol{E}(t), \Delta \boldsymbol{L}(t) \rangle_{\boldsymbol{H}}}{|\Delta \boldsymbol{E}(t)|_{\boldsymbol{H}} |\Delta \boldsymbol{L}(t)|_{\boldsymbol{H}}} = 1 \tag{15}$$

and the distribution divergence satisfies

$$\text{KL}(P_t \| Q_t) = 0, \tag{16}$$

then DRA is monotonically increasing with $\cos(\boldsymbol{E}(t), \boldsymbol{L}(t))$.

**Proof:** Under these conditions, the trajectory coherence term simplifies to 1, and the exponential penalty term becomes

$$e^{-\alpha \cdot 0} = 1. \tag{17}$$

Substituting into the DRA formulation gives

$$\text{DRA} = \frac{1}{Z_T} \sum_{t=1}^{T} \omega(t) \cdot \cos(\boldsymbol{E}(t), \boldsymbol{L}(t)), \tag{18}$$

where the $\ell_2$-normalization factor is

$$Z_T = \sqrt{\sum_{t=1}^{T}\left[\omega(t) \cdot \cos(\boldsymbol{E}(t), \boldsymbol{L}(t))\right]^2 + \sum_{t=1}^{T}\omega(t)^2}. \tag{19}$$

Since $\omega(t) \propto \text{Gamma}(t; \beta, 1)$ and $\sum_{t=1}^{T}\omega(t) = 1$, $Z_T$ is a positive quantity. Under the proposition's assumption, we treat $Z_T$ as independent of the monotonic variation of $\cos(\boldsymbol{E}(t), \boldsymbol{L}(t))$. Let $K = \frac{1}{Z_T}$, then

$$\text{DRA} = K \cdot \sum_{t=1}^{T}\omega(t) \cdot \cos(\boldsymbol{E}(t), \boldsymbol{L}(t)). \tag{20}$$

For any two sets $\{\cos(\boldsymbol{E}(t), \boldsymbol{L}(t))\}_{t=1}^{T}$ and $\{\cos'(\boldsymbol{E}(t), \boldsymbol{L}(t))\}_{t=1}^{T}$ with $\cos'(\boldsymbol{E}(t), \boldsymbol{L}(t)) \geq \cos(\boldsymbol{E}(t), \boldsymbol{L}(t))$, we have

$$\sum_{t=1}^{T}\omega(t) \cdot \cos'(\boldsymbol{E}(t), \boldsymbol{L}(t)) \geq \sum_{t=1}^{T}\omega(t) \cdot \cos(\boldsymbol{E}(t), \boldsymbol{L}(t)). \tag{21}$$

Since $K > 0$, this implies $\text{DRA}' \geq \text{DRA}$, completing the proof.

2. Proof of Robustness to Bounded Noise

**Proposition:** For bounded additive noise $\delta\boldsymbol{E}(t)$ with $|\delta\boldsymbol{E}(t)|_{\boldsymbol{H}} \leq \delta_{\max}$, the difference between noisy DRA ($\text{DRA}_\delta$) and original DRA is bounded by a constant proportional to $\delta_{\max}$.

**Proof:** Let

$$\boldsymbol{E}_\delta(t) = \boldsymbol{E}(t) + \delta\boldsymbol{E}(t), \quad \Delta\boldsymbol{E}_\delta(t) = \Delta\boldsymbol{E}(t) + \delta\Delta\boldsymbol{E}(t), \tag{22}$$

where $|\delta\Delta\boldsymbol{E}(t)|_{\boldsymbol{H}} \leq 2\delta_{\max}$. Then each term in DRA satisfies

$$|\text{DRA}_\delta - \text{DRA}| \leq C\,\delta_{\max}, \tag{23}$$

for some constant $C$, proving robustness.

3. Proof of Normalization ($\text{DRA} \in [0, 1]$)

**Proposition:** DRA is constrained within $[0, 1]$ by the normalization scheme.

**Proof:** Define the per-step alignment score as

$$x_t = \cos(\boldsymbol{E}(t), \boldsymbol{L}(t)) \cdot \frac{\langle \Delta\boldsymbol{E}(t), \Delta\boldsymbol{L}(t)\rangle_{\boldsymbol{H}}}{|\Delta\boldsymbol{E}(t)|_{\boldsymbol{H}}|\Delta\boldsymbol{L}(t)|_{\boldsymbol{H}} + \epsilon} \cdot e^{-\alpha \cdot \text{KL}(P_t \| Q_t)} \in [0, 1]. \tag{24}$$

Then with normalized weights $\omega(t) \geq 0$, $\sum_{t=1}^{T}\omega(t) = 1$, the DRA is defined as

$$\text{DRA} = \frac{1}{Z_T}\sum_{t=1}^{T}\omega(t)\,x_t. \tag{25}$$

Since each $x_t \in [0, 1]$ and the weights form a convex combination, it follows directly that

$$\text{DRA} \in [0, 1], \tag{26}$$

achieving 1 for perfect alignment and 0 for no alignment.

Overall, the proposed DRA metric provides a comprehensive measure of EEG-LLM trajectory alignment by integrating feature similarity, temporal coherence, and distributional consistency, thereby ensuring that larger DRA values directly reflect stronger alignment across both spatial and dynamic dimensions.

### A.4 DETAILS ON THE LLMS

We provide comprehensive details of the 16 LLMs in Table 3. All the experiments were implemented via the Transformers and PyTorch libraries. Model training and evaluation were performed on an NVIDIA A100 GPU with 80 GB of RAM.

Table 3: Large language models (LLMs) used in this study.

| Year | Parameter Size | Layers | Model Name |
|------|----------------|--------|------------|
| 2023 | 7B | 32 | Llama-2-7b-hf |
| 2023 | 7B | 32 | Llama-2-7b-chat-hf |
| 2024 | 8B | 40 | Meta-Llama-3-8B |
| 2024 | 8B | 40 | Meta-Llama-3-8B-Instruct |
| 2024 | 7B | 32 | Qwen2.5-7B |
| 2024 | 7B | 32 | Qwen2.5-7B-Instruct |
| 2023 | 7B | 32 | Mistral-7B-v0.1 |
| 2023 | 7B | 32 | Mistral-7B-Instruct-v0.3 |
| 2024 | 7B | 32 | gemma-7b |
| 2024 | 7B | 32 | gemma-7b-it |
| 2023 | 7B | 32 | Falcon3-7B-Base |
| 2023 | 7B | 32 | Falcon3-7B-Instruct |
| 2023 | 9B | 36 | Yi-1.5-9B |
| 2023 | 9B | 36 | Yi-1.5-9B-Chat |
| 2024 | 7B | 32 | deepseek-llm-7b-base |
| 2025 | 7B | 32 | DeepSeek-R1-Distill-Qwen-7B |

## A.5 More results on the ZuCo 2.0 dataset

We have performed the additional validation on the ZuCo 2.0 dataset(Hollenstein et al., 2020). Table A.5 presents the representational similarity results on the ZuCo 2.0 dataset. Notably, Mistral-7B-Instruct-v0.1 achieved the best MSE of 0.9595 and a Pearson correlation of 0.2499, whereas Falcon3-7B-Instruct had the highest correlation at 0.2586. In terms of structural similarity, Mistral-7B-Instruct-v0.1 led in CKA with 0.4194, followed by Yi-1.5-9B-Chat at 0.4057. The instruction-tuning effect was consistently observed, as instruction-tuned variants generally outperformed their base counterparts across most model families, reinforcing that alignment training enhances neural representational alignment.

Table 4: Sentence-level alignment results between LLM representations and brain signals on ZuCo 2.0 dataset. Best results per column are in **bold**.

| Model | MSE ↓ | $r$ ↑ | RSA ↑ | CKA ↑ |
|-------|-------|-------|-------|-------|
| Llama-2-7b-hf | 1.0648 | 0.1805 | 0.1534 | **0.3927** |
| Llama-2-7b-chat-hf | 1.0294 | 0.1931 | 0.1693 | 0.3846 |
| Meta-Llama-3-8B | 1.0632 | 0.2047 | 0.1377 | 0.3553 |
| Meta-Llama-3-8B-Instruct | 1.0611 | 0.2019 | 0.1602 | 0.3848 |
| Qwen2.5-7B | 1.1922 | 0.1535 | 0.1309 | 0.2616 |
| Qwen2.5-7B-Instruct | 1.1807 | 0.1545 | 0.1330 | 0.2748 |
| Mistral-7B-v0.1 | 0.9942 | 0.2419 | **0.2527** | 0.4019 |
| Mistral-7B-Instruct-v0.1 | 0.9595 | 0.2499 | 0.2440 | **0.4194** |
| gemma-7b | 1.0946 | 0.2169 | 0.2211 | 0.3791 |
| gemma-7b-it | 1.0449 | 0.2256 | 0.2365 | 0.3804 |
| Falcon3-7B-Base | 1.3083 | 0.2517 | 0.2201 | 0.0854 |
| Falcon3-7B-Instruct | 1.3257 | **0.2586** | 0.2143 | 0.0654 |
| Yi-1.5-9B | 1.0036 | 0.2321 | 0.2373 | 0.3978 |
| Yi-1.5-9B-Chat | **0.9966** | 0.2353 | 0.2424 | 0.4057 |
| deepseek-llm-7b-base | 0.8470 | 0.2324 | 0.1896 | 0.1041 |
| DeepSeek-R1-Distill-Qwen-7B | 0.8721 | 0.1960 | 0.1534 | 0.0708 |

### A.6 MORE RESULTS ON THE CHINESEEEG DATASET

As shown in Figure 7, the correlation between the temporal dynamics of EEG-LLM topomaps and the brain regions involved in language comprehension highlights distinct patterns. In the Chinese language comprehension task, the positive correlation in the prefrontal region at 0 ms reflects the initiation of early semantic representation in language processing, which is consistent with the prefrontal cortex's function in the initial semantic encoding of language comprehension. The significant positive correlation in the parietal region at 100 ms reflects the role of the parietal lobe in language information integration and attention regulation, facilitating the rapid recognition and meaning extraction of Chinese words. The complex correlation distribution in multiple regions at 200 ms corresponds to the interaction stage of semantics and syntax in language comprehension, where the coordination and competition of different brain regions are manifested. The expansion of negative correlation regions after 300 ms and the negative correlation in the bilateral temporal regions at 400 ms are related to the temporal lobe's function in late language integration and context-dependent semantic processing. These findings show that there are spatiotemporal coupling differences between EEG activity and LLM in different stages of Chinese language comprehension (from semantic initiation to contextual integration), providing experimental support from the brain region and temporal dimensions for analysing the similarities and differences between the neural mechanism of human Chinese language comprehension and large language models.

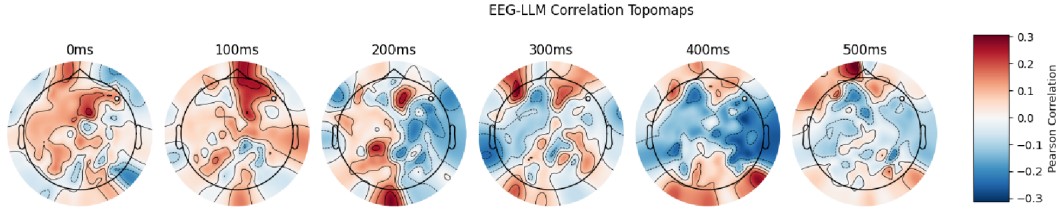

Figure 7: EEG-LLM correlation topomaps on the ChineseEEG dataset.

**Uncertainty Dynamics.** In the uncertainty entropy value (Figure 8 (A)), LLM begins with high entropy, which decreases sharply and continuously, signifying a gradual mitigation of uncertainty during processing. In contrast, EEGs exhibit relatively stable fluctuations, reflecting the brain's steady and ongoing information integration. The vertical dashed lines mark distinct change points, emphasizing the divergent strategies for handling uncertainty between artificial and biological language processing systems.

**Confidence Dynamics.** In the confidence value (Figure 8 (B)), the LLM maintains near-zero confidence for most layers before a sudden spike at Layer 30, indicating delayed, stage-final confidence consolidation. EEG, however, shows frequent peaks and troughs, suggesting real-time, dynamic confidence adjustments during linguistic processing. This contrast highlights the difference between the brain's adaptive confidence regulation and the model's delayed, stepwise confidence buildup.

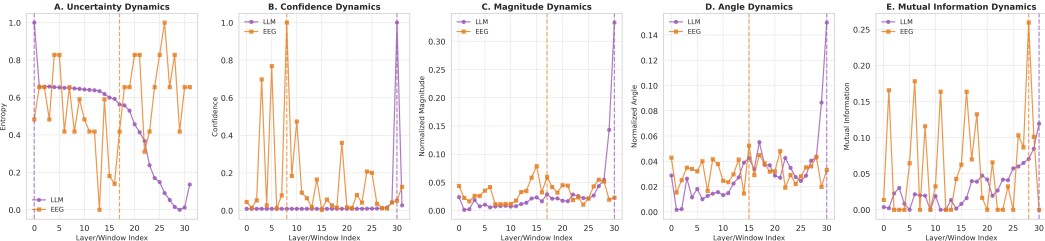

Figure 8: Temporal and dynamic comparisons between EEGs and LLMs on the ChineseEEG dataset. (A) Magnitude patterns. (B) Angle patterns. (C) Uncertainty dynamics. (D) Confidence dynamics. (E) MI dynamics.

**Magnitude Patterns.** As shown in (Figure 8 (C)), the magnitude features reveal strikingly different temporal dynamics between EEGs and LLMs. EEGs show continuous fluctuations with gradual

changes, reflecting the brain's rapid, distributed, and iterative neural processing in magnitude-related linguistic computations. In contrast, LLMs remain largely stable before a sharp surge at step 30, resembling a "silent analysis followed by late integration." This highlights a divergence between the brain's stepwise recalibration and the model's delayed, stage–end consolidation in magnitude feature processing.

**Angle Patterns.** In Figure 8 (D), the angle features further underscore complementary rhythms. EEGs display irregular fluctuations with multiple small peaks, which is consistent with ongoing neural reorientation in angle-related semantic processing. However, LLMs rise gradually and spike only at step 30, suggesting sequential and hierarchical adjustments. These results capture a contrast between the brain's real-time semantic calibration and the model's "delayed burst" processing in angle feature dynamics.

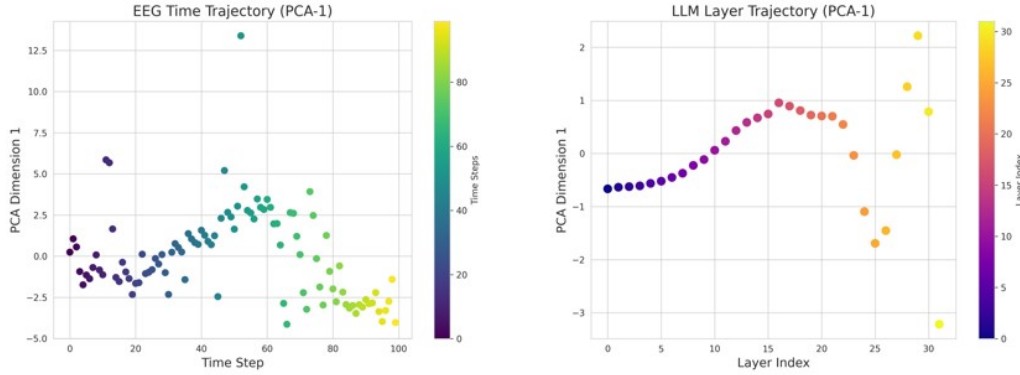

Figure 9: Left: PCA-1 trajectory of EEG responses across time steps, colored by time stage. Right: PCA-1 trajectory of LLM layer activations across layer indices, colored by layer depth.

To analyse the representational dynamics, we visualized the first principal component (PCA-1) of the EEG responses and large language model (LLM) layer activations (Figure 9. The left panel depicts the EEG time trajectory: PCA-1 clearly progresses across time steps, with distinct clusters colored by time stage, indicating evolving representations as the task unfolds. The right panel shows the LLM layer trajectory: PCA-1 forms a smooth, structured curve across layer indices, with colors encoding layer depth. Notably, the LLM's representational trajectory mirrors key trends in the EEG trajectory—both display systematic shifts that suggest hierarchical or sequential representational processing. This alignment implies that the LLM captures temporal or task-dependent representational dynamics analogous to those in human EEG, supporting the model's capacity to emulate representational patterns.

