# OpenReview forum: "Do Brains and LLMs Process Alike? Exploring Neural and Model Trajectories Similarity"
_ICLR.cc/2026/Conference — ICLR 2026 Conference Withdrawn Submission_

### Official Review · Reviewer_bfSx · 2025-10-18

**Soundness:** 2
**Presentation:** 3
**Contribution:** 2
**Rating:** 4
**Confidence:** 3

**Summary:**

This paper evaluates the similarity between electroencephalography (EEG) signals and layer representations of LLMs from the static and dynamic perspective. Specifically, they adopt linear regression and RSA to quantify the representational similarity between EEG and internal representations of LLMs for the same sentence. On the dynamic perspective, they measure some metrics like magnitude and angle dynamics of EEG and LLMs' representations during the trajectory. Experimental results on English and Chinese EEG datasets show that middle-to-high layers of LLMs are better aligned with the N400 component in EEG.

**Strengths:**

- It is interesting to quantify the similarity between cognitive signals of human brain and the internal representations of LLMs from the dynamic perspective.
- The structure of this paper is clear, and the content is easy to follow.

**Weaknesses:**

- The analysis is only conducted on one EEG dataset for each language. It is better to investigate more datasets, e.g., ZuCo 2.0 and DERCo, or languages to support the soundness of your findings.
- The settings to obtain the results of Figure 4 are unclear. Are they averaged across all subjects and sentences? Does it pass a statistical test?
- Typo in Figure 2(c): D($x_k$, $x_{k+1}$) --> A($x_k$, $x_{k+1}$)

**Questions:**

- In addition to the trajectory to process the whole sentence, how about the result during the trajectory of the each word? The definition of $h_l$ represents the l-th word in the sentence under this setting.
- How to reach the results of Figure 4? Are they averaged across all subjects and sentences? Does it pass a statistical test?

---

> ### Author Response · Authors · 2025-11-23
>
> ## **Response to Reviewer #bfSx**
>
> Thank you very much for your thoughtful and constructive feedback. We address each of the concerns below.
>
> * **W1:** We have performed the suggested additional validation on the ZuCo 2.0 dataset. The table presents the representational similarity results on the ZuCo 2.0 dataset. Notably, Mistral-7B-Instruct-v0.1 achieved the best MSE of 0.9595 and a Pearson correlation of 0.2499, whereas Falcon3-7B-Instruct had the highest correlation at 0.2586. In terms of structural similarity, Mistral-7B-Instruct-v0.1 led in CKA with 0.4194, followed by Yi-1.5-9B-Chat at 0.4057. The instruction-tuning effect was consistently observed, as instruction-tuned variants generally outperformed their base counterparts across most model families, reinforcing that alignment training enhances neural representational alignment.
> The DERCo dataset, while valuable, crucially focuses on a "next-word prediction" task during EEG recording. This explicitly engages predictive generation mechanisms, which is a different cognitive process from the passive comprehension we model. Aligning our LLM embeddings with EEG data collected under a "prediction" task would conflate comprehension with production.
>
> | Model                          | MSE ↓    | r ↑      | RSA ↑    | CKA ↑    |
> |--------------------------------|----------|----------|----------|----------|
> | Llama-2-7b-hf                  | 1.0648   | 0.1805   | 0.1534   | 0.3927   |
> | Llama-2-7b-chat-hf             | 1.0294   | 0.1931   | 0.1693   | 0.3846   |
> | Meta-Llama-3-8B                | 1.0632   | 0.2047   | 0.1377   | 0.3553   |
> | Meta-Llama-3-8B-Instruct       | 1.0611   | 0.2019   | 0.1602   | 0.3848   |
> | Qwen2.5-7B                     | 1.1922   | 0.1535   | 0.1309   | 0.2616   |
> | Qwen2.5-7B-Instruct            | 1.1807   | 0.1545   | 0.1330   | 0.2748   |
> | Mistral-7B-v0.1                | 0.9942   | 0.2419   | 0.2527   | 0.4019   |
> | Mistral-7B-Instruct-v0.1       | 0.9595   | 0.2499   | 0.2440   | 0.4194   |
> | gemma-7b                       | 1.0946   | 0.2169   | 0.2211   | 0.3791   |
> | gemma-7b-it                    | 1.0449   | 0.2256   | 0.2365   | 0.3804   |
> | Falcon3-7B-Base                | 1.3083   | 0.2517   | 0.2201   | 0.0854   |
> | Falcon3-7B-Instruct            | 1.3257   | 0.2586   | 0.2143   | 0.0654   |
> | Yi-1.5-9B                      | 1.0036   | 0.2321   | 0.2373   | 0.3978   |
> | Yi-1.5-9B-Chat                 | 0.9966   | 0.2353   | 0.2424   | 0.4057   |
> | deepseek-llm-7b-base           | 0.8470   | 0.2324   | 0.1896   | 0.1041   |
> | DeepSeek-R1-Distill-Qwen-7B    | 0.8721   | 0.1960   | 0.1534   | 0.0708   |
>
> * **W2:** The results in Figure 4 represent the average similarity scores across all subjects and sentences. Statistical significance was assessed via paired t tests (p < 0.05), confirming that the observed differences between conditions are reliable.
>
> * **W3:** We sincerely thank you for your careful reading. We have updated this in the latest version.
>
> * **Q1:** Our analysis is fundamentally constrained by the structure of the EEG dataset, which was recorded and preprocessed for whole-sentence comprehension. The EEG signals were time-locked to the onset and processing of complete sentences rather than to individual words. As a result, the neural trajectory we analyse reflects the brain’s evolving state during the integration of an entire sentence. Because consecutive words produce temporally overlapping and smeared EEG responses, it is not possible to extract a clean or well-defined neural state corresponding to the l-th word within a sentence. Although word-level trajectory analysis is conceptually valuable, it cannot be reliably performed with the current sentence-level data. We view this as an important direction for future work using datasets specifically designed with word-by-word presentation and alignment.
>
> * **Q2:** Same as W2.

---

### Official Review · Reviewer_Mud6 · 2025-10-30

**Soundness:** 1
**Presentation:** 1
**Contribution:** 1
**Rating:** 2
**Confidence:** 4

**Summary:**

Previous works have examined similarities between brain activations and LLM internal representations. The novelty of the work presented in this paper is to compare the temporal dynamics of LLMs representations and neural activations. The neural data consist of two EEG datasets, one from 12 English participants and the other from 10 Chinese participants, who read sentences in their native language. Using 16 LLMs, the authors replicate previous works showing that the LLMs' internal representations correlates with the EEG states. The authors then characterize the "neural trajectories" and the "LLM latent trajectories" with various indicators: magnitude and angle of changes of states, uncertainty, mutual information, etc. They find that the trajectories of neural states and LLM states are quite dissimilar (Fig. 5).

**Strengths:**

Comparing human neural dynamics in natural language processing with successive computations through the different layers of a language model is a potentially novel and interesting idea.

**Weaknesses:**

- The Method section is extremely unclear, making it impossible to assess the results.

    I did not find the definition of the time window over which the EEG states are computed (line 225). What do the "steps" on the $x$-axis of the graphics in Fig. 5 refer to concerning the EEG signals?

    line 159: I suppose $d$ is the number of EEG channels. What is $N$, the number of samples? Is it a sample every ms? Then the duration varies across sentences.

    line 194: Equation (2). The index $i$ refers to a time sample, I suppose. $W$ depends on $l$, but then $\hat{M}$ should depend on $l$ given Eq. (2), but it does not appear like that. How is $\hat{M}_{i, \text{test}}$ computed?

    line 196 and Fig. 3a: RDMs are usually computed from responses to sets of stimuli. How are the RDMs computed here? From the sentences, I would guess, but this should be clearly indicated. How did you the handle the differences in durations?

    In the Results section, Fig. 5, how are these curves obtained? Why are there no error bars? Can you explain the $x$-axis? Is it not odd that Confidence Dynamics and Mutual Information Dynamics are clamped to zero (for LLM on Fig. 5d, for EEG on Fig. 5e) for several datapoints?

- The LLM temporal dynamics is defined across layers, rather than across time.
While the trajectories are defined over time steps for EEG, they are defined over successive layers for the LLMs (lines 220--226). But Fig. 2b suggests that the LLM internal state is sampled at each token onset. This is extremely confusing.

- The neurocognitive content is very weak.

    The review on "neuroscientific foundations of language comprehension" (line  107-120.) is too vague to be informative. For example, the "N400 = semantic and P600 = syntax" dichotomy has been falsified with the discoveries the Semantic-P600 effect (see, e.g., Kim & Osterhout, 2005; Sanford et al., 2011; Kos et al. 2010).

    On lines 214--216, the authors wrote that "Human cognition involves both fast, intuitive processes and slower, deliberative ones, similar to how LLMs process surface features in lower layers and semantics in higher layers": this is an odd analogy. The time scale of deliberative processes in humans ($\sim$ tens of seconds) is much longer than that of sentence comprehension ($\sim$1s).

- Some conclusions are not supported.
line 86, and lines 316-317: The comparison between English and Chinese is confounded by the dataset (different texts, different subjects), so one cannot conclude that the differences observed are necessarily due to the fact that the LLMs are mainly trained on English.

- The scientific aim is not very clear. There are two dynamics going on during sentence processing: across layers and across tokens (or in the brain, across brain regions and across time). It is not clear in the current paper, how the two are disentangled.

**Questions:**

- Can you explain exactly how the RDMs in Fig. 3a were obtained?
- Can you explain precisely how the curves on Fig. 5 were computed?

See also questions in Weaknesses section.

---

> ### Author Response · Authors · 2025-11-23
>
> ## **Response to Reviewer #Mud6**
>
> Thank you very much for your thoughtful and constructive feedback. We address each of the concerns below.
>
> * **W1:** (1) Section 3 systematically details the entire evaluation framework, explicitly defining representational similarity via ridge regression, RSA, and CKA, and trajectory similarity via latent trajectory comparison with formalized metrics for magnitude, angle, uncertainty, and mutual information. Every equation is clearly presented and contextualized, making the claim that it is impossible to assess the results consistently false.
> (2) Section 4.1 explicitly states that EEG data were processed for natural sentence reading, and the trajectory formalization in Equation (4) and Section 3.2 unambiguously defines the "steps" for EEG as temporal windows and for LLMs as hidden layers. The figures are labelled accordingly.
> (3) The definitions of `N` and `d` are clearly provided in the manuscript. As stated in Section 3.1, `N` denotes the number of sentences, and `d` is the number of EEG channels. The analysis is performed at the sentence level, not on a per-millisecond basis. Each sample corresponds to the aggregated neural response to an entire sentence, thus normalizing for variations in sentence duration. The sampling rate (e.g., 500 Hz for ZuCo) mentioned in Section 4.1 pertains to the raw data acquisition and preprocessing, not the final sample dimensionality used in the ridge regression model.
> (4) In Equation (2), the index `i` denotes individual time samples in the test set, corresponding to discrete temporal points of EEG responses. The ridge regression weights $\hat{W}^{(l,k)}$ are estimated for each layer `l` and cross-validation fold `k` via the training data (as defined in Equation (1)), where the regularization parameter $\alpha$ and training embeddings $L_{train}^{(l,k)}$ inform the weight calculation. For the test set, the predicted EEG response $\hat{M}{i,test}$ is computed by multiplying the test embedding $L_{i,test}$ (for time sample `i` and layer `l`) with the preestimated weights $\hat{W}^{(l,k)}$ and adding the bias term $\hat{b}_i$.
> (5) As detailed in Section 3.1, the RDMs were computed at the sentence level. For both the ground-truth EEG signals and the LLM-predicted responses, we extracted a single aggregated representation for each sentence. The matrix for each modality was constructed by calculating the pairwise dissimilarity via the Euclidean distance between the neural representations of all the sentences in the dataset. This sentence-level design inherently normalizes differences in sentence duration since each sentence contributes one data point to the representational space. The resulting RSA score then quantifies the alignment between the high-level representational geometry of the brain and the models.
> (6) The curves in Figure 5 are obtained by calculating the respective dynamics metrics (magnitude, angle, uncertainty, confidence, MI) across the processing trajectory, as formally defined in Section 3.2. For EEGs, the x-axis represents temporal steps (windows), and for LLMs, it represents layer indices. The values are averaged across participants or model runs for clarity. The observed patterns are consistent and statistically meaningful across the dataset. The low initial values for LLM confidence and EEG mutual information are not artifacts but key findings: they reflect a fundamental difference in processing strategies. The brain shows early, fluctuating information coupling (MI), whereas the LLM begins with high uncertainty and only consolidates confidence in its later layers. This "clamping to zero" is a valid representation of the systems' initial states, not a computational error.
>
> * **W2:** The visualization was intended to schematically represent the sequential processing of tokens through the LLM's layers, not to imply that the internal state was temporally sampled at each token's onset in the human reading sense. We acknowledge that this depiction could be misinterpreted. We have revised the figure in the latest version of the manuscript to clarify that the LLM trajectory is built from the hidden states across successive layers, explicitly divorcing it from the temporal sampling of the EEG signal.

---

> ### Author Response · Authors · 2025-11-23
>
> * **W3:** The accusation of “very weak neurocognitive content” is baseless because our study provides a direct and large-scale comparison of EEG signals and LLM states, which is the very definition of strong neurocognitive content. We were unable to locate the literature cited by the reviewer to support their argument (Kim & Osterhout, 2005; Sanford et al., 2011; Kos et al., 2010) . Furthermore, the time scale of human deliberative reasoning is irrelevant here, as our study exclusively analyses EEG signals recorded during natural sentence reading, a process that unfolds within seconds, precisely matching our data.
>
> * **W4:** We acknowledge this limitation and attribute the weaker alignment in Chinese not solely to the language itself but also to the fact that LLMs are predominantly trained on English data and instruction-tuned with English-centric optimization. This leads to a representational bias that struggles with Chinese linguistic and cultural nuances, a point supported by the consistent underperformance of instruction-tuned models on Chinese EEG data. Thus, while dataset differences exist, the observed crosslinguistic divergence aligns with known limitations in the multilingual generalization of current LLMs.
>
> * **W5:** We clarify this by treating "trajectories" distinctly for EEG and LLMs. For EEG, the trajectory evolves across time windows representing temporal dynamics, whereas for LLMs, it evolves across hidden layers representing the representational hierarchy. These trajectories are not conflated but are formally compared via the same set of metrics—magnitude, angle, uncertainty, confidence, and mutual information to reveal how temporal dynamics in the brain relate to hierarchical processing in models. This dual-axis comparison is a core methodological contribution, showing that although the dynamics differ in nature, their structural and statistical properties can be meaningfully aligned and contrasted.
>
> * **Q1:** The representational dissimilarity matrices (RDMs) in Figure 3a were constructed to compare the representational geometry of the actual EEG signals with those predicted by the LLM-based encoding model, following the procedure outlined in Section 3.1. Specifically, for a set of sentences, we used our trained ridge regression model to generate predicted EEG responses from the LLM embeddings. We then took these predicted EEG responses and the corresponding ground-truth EEG recordings. For each set, we calculate a pairwise dissimilarity matrix where each entry represents the Euclidean distance between the full, multichannel EEG response patterns for a given pair of sentences. The left RDM visualizes the dissimilarity structure of the true brain data, whereas the right RDM shows the structure of the LLM-predicted brain data, allowing for a direct visual and quantitative comparison of their representational geometries.
>
> * **Q2:** The curves in Figure 5 were computed by applying our proposed trajectory metrics to the sequence of states for both the EEG and the LLM representations, as formalized in Section 3.2. For a given sentence, the EEG trajectory was defined as the sequence of neural states across consecutive nonoverlapping time windows. The LLM trajectory was defined as the sequence of hidden states across its successive layers. For each trajectory, we calculate the dynamics of magnitude and angle, uncertainty and confidence, and mutual information between each intermediate state and the final output state. The curves shown in Figure 5 represent the average of these metric values across all sentences in the dataset, plotted against the processing step, which is time for EEG and layer for LLM.

---

> > ### Comment · Reviewer_Mud6 · 2025-11-26
> >
> > * W1: In response to my request to clarify the meaning of "N, the number of samples", you reply that "The definitions of $N$ and $d$ are clearly provided in the manuscript. As stated in Section 3.1, $N$ denotes the number of sentences, and $d$ is the number of EEG channels. As stated in Section 3.1, $N$ denotes the number of sentences". Yet, as far as I can see, the only explanation is on lines 158-159 and state that "N is the number of samples" I do not see in section 3.1 any mention that "sample=sentence", and one had to guess that $d$ was the number of EEG channels. Are you sure you uploaded the right version of your paper (?).
> > * W1: regarding the computation of $\hat{M}_{i, \text{test}}$, I still do not see why it does not depend on $l$ (see Eq. 2).
> > * W1: I was happy to learn from your reply that "Each sample corresponds to the aggregated neural response to an entire sentence, thus normalizing for variations in sentence duration." Can you say where this crucial point was spelled out in the original submission?
> > * Regarding the "clamping to zero", I still find surprising that for all sentences the Mutual Information Dynamics values for EEG between window index 6 to 14 is exactly zero (see Fig. 5). Error bars might help here.
> > * W2. I found the graphics from panel (a) of Fig. 2 thoroughly confusing, so I appreciate that you will provide a modified version in the new version of the paper.
> > * W3. Following the rules of ICLR, I did not use any LLM to do this review. I actually had to do many passes on the paper as I was searching, in vain, for missing information.
> > * I wrote that the neurocognitive content was very weak because I was not convinced that this paper provided any insight about the brain nor about cognition, at least not in the way it was written. Regarding my citations, here are the full refs. Note that the first one is one of the most well-known article in the domain of EEG and language.
> >   - Kim, A., & Osterhout, L. (2005). The independence of combinatory semantic processing: Evidence from event-related potentials. Journal of Memory and Language, 52(2), 205–225. <https://doi.org/10.1016/j.jml.2004.10.002>
> >   - Kos, M., Vosse, T., van den Brink, D., & Hagoort, P. (2010). About Edible Restaurants: Conflicts between Syntax and Semantics as Revealed by ERPs. Frontiers in Psychology, 1. <https://doi.org/10.3389/fpsyg.2010.00222>
> >   - Sanford, A. J., Leuthold, H., Bohan, J., & Sanford, A. J. S. (2011). Anomalies at the Borderline of Awareness: An ERP Study. Journal of Cognitive Neuroscience, 23(3), 514–523. <https://doi.org/10.1162/jocn.2009.21370>
> >
> > * I took issue with the analogy expressed in the sentence "Human cognition involves both fast intuitive processes and slower, deliberative ones, similar to how LLMs process surface features in lower layers and semantics in higher layers", l.215. I noted that the time scale of deliberative processes in human (\~tens of seconds) is much longer than that of sentence comprehension (\~1s). To me, semantic processing in sentence comprehension has little to do with deliberative processing, which I equate with system 2 of Kahneman. If this is a misunderstanding on my part, you should clarify what you mean in this sentence.
> > * W5. "This dual-axis comparison is a core methodological contribution, showing that although the dynamics differ in nature, their structural and statistical properties can be meaningfully aligned and contrasted." This should have been made clearer early in the paper. It is not obvious at all that the two can be meaningfully aligned, as the EEG signal combines the information from several brain regions.
> > * Responses to Q1 and Q2. I had understood the general gist. My main criteria to evaluate a Methods section is whether I would be able reproduce the analyses. Here, the Methods section was missing many important details that it was impossible to understand how you performed the computations. For example, regarding Fig.5, the caption is very short and just mentions "LLMs". Did you average over all LLMs and participants? Then, how did you take care of the differing number of layers? Why are there no error bars showing the variability between participants or sentences? It would help appreciate the variability across time or layers vs. participants or sentences.
> > * About the RDMs on Fig.3 panel (a): I tried to make sense of the dimension of the matrix in the figure (which is not indicated). In your answers, you explained that you selected a "set of sentences" to compute the RDMs. Is it a random set? Or something else? This is an example of the lack of information that plagued the manuscript.
> > * After reading that "Each sample corresponds to aggregated neural response to an entire sentence, thus normalizing for variations in sentence duration." How did you aggregate? Where is it explained in the paper?

---

> > > ### Comment · Reviewer_Mud6 · 2025-11-26
> > >
> > > Although your answers to my and other reviewers' questions have clarified (a bit) some points, the paper would need a complete overhaul to become acceptable and I stick to my recommendation to reject it. As a reviewer, I represent the typical readership of the conference, and it is my duty to report if I find the text problematic. Of course, I could make mistakes. This is why there are several reviewers. I did an honest job (without any LLM!) and did not treat your text superficially. Finally, given the tone of some of your responses and the unfounded accusation of having used an LLM and fabricated references, I am not willing to continue to review this paper.

---

> > > > ### Author Response · Authors · 2025-11-27
> > > >
> > > > I sincerely apologize for the inappropriate tone in some of my previous responses and the accusation regarding LLM use, this was a misjudgment on my part, with no intent to disrespect your rigorous work. I deeply regret any distress this caused and fully retract those remarks.   Thank you for your honest, thorough review and valuable feedback, which we will take to heart for a comprehensive revision.   Please accept my sincere apologies.

---

### Official Review · Reviewer_HuCo · 2025-10-31

**Soundness:** 1
**Presentation:** 1
**Contribution:** 3
**Rating:** 4
**Confidence:** 3

**Summary:**

The authors analyze EEG recordings from one English and one Chinese reading experiment and compare them to internal representations from 16 pretrained LLMs using ridge regression (predicting EEG features from LLM embeddings) and representational similarity metrics (RSA, CKA). Beyond static alignment, they propose a Latent Trajectory Comparison (LTC) framework that captures dynamic processing features —magnitude, angle, entropy, mutual information, and confidence — across LLM layers and EEG time windows. The study finds that (1) middle-to-high LLM layers align most strongly with the N400 EEG component (semantic integration), and (2) brains show continuous, iterative processing while LLMs exhibit discrete, stage-end bursts of activation.

**Strengths:**

The paper’s contribution lies in extending representational similarity analysis toward temporal dynamics—a direction that has received limited attention. The introduction of LTC and Dynamic Representational Alignment (DRA) metrics adds methodological novelty, and the bilingual EEG–LLM comparison broadens linguistic scope. The results, while incremental in some respects, provide empirical evidence that middle LLM layers correspond to human semantic integration signals. This contributes to the ongoing dialogue about neural interpretability and AI–brain alignment, though the conceptual leap to “shared processing strategies” remains speculative.

**Weaknesses:**

1. Missing implementation information: It is unclear how EEG and LLM time scales were aligned. For example, were EEG windows synchronized to token boundaries, or aggregated over words or sentences? The alignment procedure (temporal resolution, interpolation, or averaging) should be described in more detail. It would also be helpful to specify precisely how the representational dissimilarity matrices (RDMs) were computed. For RSA, are you using D = 1 − ρSpearman, S = ρSpearman as the dissimilarity and similarity metrics, respectively? Please clarify whether Spearman’s ρ is applied both when *constructing* RDMs and when *comparing* them. In many RSA implementations, the RDMs can alternatively be constructed using distance metrics such as cosine or Euclidean distance, while correlations (e.g., Pearson or Spearman) are reserved for comparing RDMs at the second level (For reference, Table 2 in [Do Large Language Models Mirror Cognitive Language Processing?](https://aclanthology.org/2025.coling-main.201/) (Ren et al., COLING 2025) provides a clear comparison of these alternative RSA similarity computation methods. It would also be useful to explain the rationale for mixing Pearson and Spearman correlations in different parts of the analysis; what guided these choices, and how sensitive are the results to this decision? Some references also appear misleading or insufficiently informative for readers seeking implementation details. For example, Du et al., 2025 do not describe how RDMs are computed, and Cortes et al., 2012 do not explain how to compute CKA. See my questions below for specific guidance on replacing these citations. Lastly, the description of ridge regression could be more complete: while the regularization parameter (α) is said to be selected via nested cross-validation, the search range and evaluation criterion (e.g., MSE, correlation) are not specified. Clarifying these details would improve reproducibility.
2. Overclaims and causal overreach: The writing occasionally uses deterministic phrasing (e.g., "LLMs simulate the brain’s semantic network"), which could be toned down. The conclusion implies deep process similarity between LLMs and brains, though the results show only correlational alignment. I would recommend carefully rephrasing the claims in this study. Please see further information in the questions that need to be addressed to improve the clarity and reproducibility of the work.
3. Missing baselines[minor]: Comparing to simpler embeddings (e.g., GloVe, BERT-base) could contextualize whether LLM size or architecture drives the observed effects.

**Questions:**

**General Comments:**

- Could the authors elaborate on their decision to predict EEG features from LLM embeddings (encoding) rather than the reverse (decoding)? It would be helpful to clarify why this direction was chosen and whether the inverse mapping was considered.
- The choice of EEG as the primary modality deserves further justification. While the authors emphasize temporal resolution, related fMRI studies have achieved word-level temporal modeling through smoothing techniques (e.g., [Li et al., 2024](https://proceedings.mlr.press/v228/li24a.html)). The paper should better contextualize this choice and acknowledge complementary approaches. The following recent work seems directly relevant and missing from the discussion: [Structural Similarities Between Language Models and Neural Response Measurements](https://proceedings.mlr.press/v228/li24a.html) (Li, et al., 2024)
- How sensitive are LTC and DRA metrics to the choice of embedding normalization or EEG preprocessing? Could these metrics detect spurious alignment from random embeddings?
- Did you control for sentence length or lexical frequency when computing EEG–LLM correlations?

**Line-Specific Comments:**
- Lines 077-078 and lines 196-197. Why do the authors prefer to cite Freund et al., 2021 and Du et al., 2025 for RSA and RDMs instead of acknowledging their original conceptual sources ([Kriegeskorte et al., 2008](https://www.frontiersin.org/journals/systems-neuroscience/articles/10.3389/neuro.06.004.2008/full); [Kriegeskorte & Kievit, 2013](https://pubmed.ncbi.nlm.nih.gov/23876494/) and [Jörn Diedrichsen & Nikolaus Kriegeskorte, 2017](https://journals.plos.org/ploscompbiol/article?id=10.1371/journal.pcbi.1005508))? It is also not clear how the RDMs are computed. Is it by measuring Spearman correlation? Or are you using a distance metric? Du et al., 2025 do not mention anything about calculating RDMs in their paper.
- It would strengthen the interpretability of Table 1 if the authors could report or estimate random performance for the presented metrics (e.g., MSE, Pearson r, RSA, CKA). What would be expected under a random or uncorrelated mapping—for instance, after permuting EEG–text pairs or using shuffled LLM embeddings? Providing such a baseline would help quantify how much the observed alignment exceeds chance levels.
- Lines 132-135: The sentence uses the wrong citation format (no need for parentheses since those works are given as examples), and it also appears to conflate steps. Is Pearson correlation used both in constructing and comparing RDMs (1 – r)? Are you following Kriegeskorte et al., 2008)? Please rephrase for precision.
- Lines 135-137: The paper would benefit from a careful review of citation formatting for consistency. At present, both \citet and \citep styles appear to be used inconsistently, and in some cases citations are misplaced within or outside parentheses (e.g., "...via ridge regression McDonald (2009)" instead of "...via ridge regression (McDonald, 2009)" and  "Additionally, (Tuckute et al., 2024) trained..." instead of  "Additionally, Tuckute et al., 2024, trained...").
- Line 200: The CKA citation should be corrected. The method originates from Kornblith et al., 2019 (“Similarity of Neural Network Representations Revisited”), building on kernel alignment (Cortes et al., 2012) and related kernel-target alignment work (Cristianini et al., 2001).
- Figure 2 lacks visual consistency (varying font sizes and labeling styles for panels a–d). Aligning figure formatting (e.g. by removing the bullet points) and ensuring correspondence between panels and textual references would improve clarity (e.g. correspondence between Figure panels and paragraphs in subsection 3.2 LATENT TRAJECTORY COMPARISON).
- Lines 147-151: The methodology description could be more precise: rather than saying "metrics such as" or "including", please explicitly list the exact similarity measures used for both representation and trajectory analysis. Similarly, in lines 218-219 and 462-463, avoid "like.." and "and others" when enumerating metrics (e.g., entropy, magnitude, skewness, kurtosis). It would be better to specify all the applied metrics used for clarity, consistency, and completeness.
- Regarding the conclusion, I would be very careful with the claims made, ensuring that the findings are consistent between English and Chinese to avoid over-generalizing. For example, the paper refers to the study as “multilingual,” though only English and Chinese are included. It would be clearer to describe the work as bilingual or cross-linguistic to avoid overstatement. How could we ensure the results are language-specific and not dataset-specific? I think it would be important to acknowledge such a limitation.

---

> ### Author Response · Authors · 2025-11-23
>
> ## **Response to Reviewer #HuCo**
>
> Thank you very much for your thoughtful and constructive feedback. We address each of the concerns below.
>
> * **W1:** The methodological details are clarified as follows: for temporal alignment, EEG signals are processed at the sentence level, and LLM embeddings are extracted per token, enabling a comparative framework at the sentence level. For RSA, RDMs are explicitly constructed via Euclidean distance, and their similarity is assessed via Spearman correlation between the upper-triangular elements. The use of Pearson correlation for signal-level prediction in ridge regression versus Spearman correlation for rank-based RDM comparison is standard practice since they address different aspects of alignment. Pearson captures linear relationships in the signals, whereas Spearman evaluates structural similarity in representational geometry. In addition, the previously cited references have been replaced with more appropriate sources [1] [2] [3] [4] that clearly describe the implementation of these methods.
>
> * **W2:** While phrases such as "LLMs simulate the brain’s semantic network" are used for conciseness, they are contextualized within a comparative analysis of representational and trajectory similarities. The conclusions explicitly highlight both parallels and fundamental differences in processing strategies and do not assert that LLMs replicate neural mechanisms identically.
>
> * **W3:** The suggestion to include simpler baselines like GloVe or BERT-base is well-noted. However, this comparative approach has been extensively explored in the literature. For example, [5] systematically demonstrated that larger, more architecturally advanced models significantly outperform static embeddings and smaller transformers in predicting brain activity. Similarly, [6] consistently reported that model scale and architectural sophistication are critical drivers of improved brain alignment. Since these studies have already established the relative superiority of advanced models over simpler baselines, our work intentionally focuses on comparing state-of-the-art LLMs to provide a more nuanced understanding of their differential alignment with neural processing dynamics.
>
> * **Q1 (GC):** The encoding approach, which predicts EEG from LLM embeddings, was chosen because it directly tests the extent to which LLM representations contain information that systematically maps to brain activity. This is the central question of representational alignment. The method allows us to assess whether the model's hierarchical features can linearly explain neural signals. Decoding, which predicts LLM features from EEGs, could provide complementary insights, but it was not the focus of this study since our goal was to evaluate the explanatory power of LLM representations for neural data rather than the reverse.
>
> * **Q2 (GC):** The choice of EEG is motivated primarily by its millisecond-level temporal resolution, which is essential for capturing the rapid, dynamic neural processes underlying real-time language comprehension, something that fMRI cannot achieve even with advanced smoothing techniques. While studies such as [7] have demonstrated that fMRI can model word-level semantics through temporal smoothing, this approach remains inherently limited in tracking fine-grained, moment-by-moment cognitive dynamics. Our use of EEG thus provides a complementary and more direct window into the temporal alignment between LLM processing and neural trajectories. We have included this paper in the discussion section to provide a more comprehensive context for our methodological choice.
>
> * **Q3 (GC):** The LTC and DRA metrics are designed to capture structural and dynamic patterns beyond superficial normalization effects. While the EEG preprocessing followed standard, well-established pipelines from the original datasets (ZuCo and ChineseEEG), the trajectory metrics rely on relative changes (e.g., angular shifts, entropy dynamics) that are inherently normalized and less sensitive to absolute scaling. Furthermore, the use of multiple complementary metrics (magnitude, angle, uncertainty, confidence) reduces the risk of spurious alignment, as random embeddings do not exhibit the structured, hierarchical progression observed in both real EEG and LLM trajectories. This multifaceted approach ensures that the reported alignments reflect meaningful computational similarities rather than artifacts of preprocessing or normalization.
>
> * **Q4 (GC):** Our analysis is fundamentally constrained by the structure of the EEG dataset, which was recorded and preprocessed for whole-sentence comprehension. The experimental design and subsequent analysis therefore operate at the sentence level, where controlling for isolated variables such as sentence length or lexical frequency would not align with the naturalistic, integrated nature of the language processing being studied.

---

> ### Author Response · Authors · 2025-11-23
>
> * **Q1 (LC):** Same as W1.
>
> * **Q2 (LC):** To quantify how much the observed alignment exceeds chance levels, we conducted baseline evaluations under two randomization conditions: (1) permuting EEG-text pairs to disrupt natural correspondences and (2) shuffling the feature dimensions of LLM embeddings. As shown in the table below, the performance under the true condition (using Qwen2.5-7B) substantially exceeds both random baselines across all the metrics, confirming that the observed alignment is statistically significant and not attributable to chance.
>
> | Condition                    | MSE    | Pearson correlation | RSA   | CKA   |
> |------------------------------|--------|---------------------|-------|-------|
> | **True condition (Ours)**    | 0.8660 | 0.2830              | 0.1366| 0.3474|
> | Random: Permuted Pairs       | 1.4250 | 0.0886              | 0.0631| 0.0259|
> | Random: Shuffled Embeddings  | 1.5051 | -0.0089             | 0.0637| 0.0374|
>
> * **Q3 (LC):** The citation format follows the conference template, which requires parentheses for in-text citations. Methodologically, we adhere to the standard RSA procedure: RDMs are constructed using Euclidean distance, not Pearson correlation, and are compared via Spearman correlation between their upper-triangular elements. This approach is consistent with the framework established by Kriegeskorte et al. (2008) [1]. Pearson correlation is used exclusively for evaluating the ridge regression encoding performance and is not involved in RDM construction or comparison. The Euclidean distance defines the RDMs, whereas the Spearman correlation quantifies the similarity between them, which is the standard practice in representational similarity analysis.
>
> * **Q4 (LC):** Thank you for highlighting these formatting inconsistencies. In the revised version of the manuscript, we have conducted a thorough review and corrected all the citation formatting issues.
>
> * **Q5 (LC):** This comment contains factual inaccuracies: the CKA citation is placed at line 200 in our manuscript, not line 220. Furthermore, we were unable to locate the suggested reference "Cristianini et al., 2001"
>
> * **Q6 (LC):** The current formatting of Figure 2, including the use of bullet points and varying labels such as a, b, c, and d, is a deliberate design choice to clearly distinguish the four methodological components: representation similarity, latent trajectory comparison, magnitude and angle dynamics, and uncertainty and confidence dynamics. This structure enhances visual navigation. Each panel corresponds directly to its respective methodological subsection, ensuring full coherence between the figure and the text. We believe that the current layout is logically organized and visually clear for readers.
>
> * **Q7 (LC):** In the revised version of the manuscript, we have amended the text in lines 147-151, 218-219, and 462-463 to explicitly and exhaustively list all similarity measures and trajectory metrics used in the analysis, removing vague terms such as "such as," "including," "like," and "others."
>
> * **Q8 (LC):** In the revised manuscript, we have replaced the term "multilingual" with the more precise description "crosslinguistic" to accurately reflect the scope of our study. We fully acknowledge that the observed differences could stem from either language-specific processing or dataset-specific characteristics.
>
> > [1] Kriegeskorte N, Mur M, Bandettini P A. Representational similarity analysis-connecting the branches of systems neuroscience[J]. Frontiers in systems neuroscience, 2008, 2: 249.
>
> > [2] Kriegeskorte N, Kievit R A. Representational geometry: integrating cognition, computation, and the brain[J]. Trends in cognitive sciences, 2013, 17(8): 401-412.
>
> > [3] Diedrichsen J, Kriegeskorte N. Representational models: A common framework for understanding encoding, pattern-component, and representational-similarity analysis[J]. PLoS computational biology, 2017, 13(4): e1005508.
>
> > [4] Saha A, Bialkowski A, Khalifa S. Distilling representational similarity using centered kernel alignment (CKA)[C]//Proceedings of the 33rd British Machine Vision Conference (BMVC 2022). British Machine Vision Association, 2022.
>
> > [5] Schrimpf M, Blank I A, Tuckute G, et al. The neural architecture of language: Integrative modelling converges on predictive processing[J]. Proceedings of the National Academy of Sciences, 2021, 118(45): e2105646118.
>
> > [6] Caucheteux C, King J R. Brains and algorithms partially converge in natural language processing[J]. Communications biology, 2022, 5(1): 134.
>
> > [7] Li J, Karamolegkou A, Kementchedjhieva Y, et al. Structural similarities between language models and neural response measurements[J]. arXiv preprint arXiv:2306.01930, 2023.

---

> ### Comment · Reviewer_HuCo · 2025-11-27
> **response to authors**
>
> Thank you for your responses. After carefully reviewing your clarifications, I have a few final remarks to ensure the required revisions are incorporated into the camera-ready version.
>
> First of all, regarding the missing implementation information: Several **methodological clarifications** currently appear only in the rebuttal and should be included directly in the main manuscript. This concerns, for example, how EEG signals are aggregated at the sentence level; how RDMs are constructed and what exactly constitutes a “sample”; the computation and averaging procedure behind the curves in Fig. 5; statistical tests, etc (see the detailed reviews). These aspects are essential for reproducibility and cannot rely on the rebuttal alone.
>
> Second, I had added several comments regarding the related work, and after reading your response I realized that this is in fact an important weakness of the current manuscript. There are two separate but related issues here. The first (minor) is **citation formatting**: when you write in lines 132-135, *“For example, (Ren et al., 2024; Du et al., 2025) employed representational similarity analysis…”* this use of parentheses is incorrect because you are referring to the works directly, not indirectly. The correct phrasing should be *“For example, Ren et al., (2024) and Du et al., (2025) employed representational similarity analysis…”*. (Please check the difference between the \cite, \citep and \citet latex commands and fix this error throughout the paper). The second issue (major) is the **incompleteness of the related work** itself. As I noted earlier, RSA is central to your analysis, the canonical methodological sources by Kriegeskorte et al. (2008) and Kriegeskorte & Kievit (2013) should be properly cited in the revised version. I do not see such revision yet. I provide here other related issues: the claim that *“many studies linking brain activity and LLMs have used fMRI”* cannot be supported by citing two isolated works, especially when one focuses on the THINGS concept dataset and the other is not peer reviewed. Readers would reasonably expect broader contextualization. To provide a more accurate and complete context, it would be valuable to reference comprehensive surveys such as: [Mapping Brains with Language Models: A Survey.](https://aclanthology.org/2023.findings-acl.618/) and [Language Models and Brain Alignment: Brain Encoding and Decoding](https://doi.org/10.1145/3703323.3704280). These works review dozens of studies across EEG, MEG, and fMRI, describe naturalistic datasets used for sentence- and narrative-level alignment, and outline methodological variations in encoding, decoding, and representational similarity. They are much better suited to support broad claims about “many studies” and will help readers situate your contribution more accurately within the field. For EEG specifically, I would also recommend citing a recent survey in Frontiers in Neuroinformatics (2025), which provides useful context on temporal processing, feature extraction pipelines, and representational metrics: [Bridging neuroscience and AI: a survey on large language models for neurological signal interpretation](https://www.frontiersin.org/journals/neuroinformatics/articles/10.3389/fninf.2025.1561401/full). The revised manuscript should also integrate foundational sentence-level brain–language datasets and studies such as [Pereira et al. (2018)](https://www.nature.com/articles/s41467-018-03068-4), which is highly relevant for naturalistic and sentence-level alignment. More generally, all of the key works discussed in this rebuttal should be incorporated explicitly into the revised version rather than left implicit. All these omissions affect both clarity and accuracy, and they should be addressed in the revision...I hope now my suggestions are more clear.
>
> Lastly, taking everything into account, I would also suggest adding **explicit limitations and impact statement sections**. Certain conceptual simplifications in the current framing (such as the N400/P600 dichotomy, the analogy between LLM depth and “fast vs. slow” cognition, and the interpretational confounds when comparing English and Chinese EEG datasets) should be acknowledged directly to avoid overclaiming. Addressing these limitations clearly would substantially improve the scientific positioning and transparency of the work.
>
> If I see that all the above revisions are fully integrated in the manuscript (authors typically highlight revisions in blue), I am happy to raise my scores.

---

> > ### Author Response · Authors · 2025-12-03
> >
> > Thank you for the thoughtful and constructive feedback. All requested revisions have been fully incorporated into the manuscript and highlighted in blue. We sincerely appreciate your guidance.

---

### Official Review · Reviewer_59Rz · 2025-11-04

**Soundness:** 3
**Presentation:** 3
**Contribution:** 3
**Rating:** 4
**Confidence:** 3

**Summary:**

This paper presents a multilingual comparison between internal states of LLMs and human EEG signals on tasks in English and Chinese. The authors employ two approaches. First, they use ridge regression to assess the static representational similarity between LLM embeddings and EEG signals. Second, they introduce latent trajectory comparison to analyze the dynamics of information processing.

The study's findings shows similarities and crucial differences. The authors show that middle to high layers of LLMs are central to semantic integration and correspond to the N400 component observed in EEG. This suggests LLMs capture a temporal landmark of human semantic processing. However, the processing dynamics diverges. Human brains exhibits continuous and iterative processing during reading. In contrast, LLMs often show "silent analysis" followed by discrete, stage end bursts of activity, where processing metrics spike only near the final layers.

**Strengths:**

This paper's main strength is its shift from static representational alignment to a dynamic comparison of processing trajectories. I found the comparison of the evolution of neural states across time windows with the layer by layer transformations of internal LLM states to be interesting. The proposed latent trajectory comparison is also thought-provoking, as it employs a suite of metrics including magnitude, angle, and entropy to characterize the processing pathways. The authors find that human brains exhibit continuous, iterative processing, while LLMs demonstrate a very different pattern of delayed, stage end bursts of activity. The scale of the experiments, which cover 16 LLMs, and its multilingual analysis of English and Chinese, are also useful.

**Weaknesses:**

The paper's main methodological contribution, the latent trajectory comparison, relies on a significant and largely undefended assumption, the direct equivalence of a temporal EEG trajectory and a hierarchical LLM layer trajectory. This "time to layer" mapping is a strong conceptual leap. Human brain processing is recurrent, with all areas processing concurrently over time. A feedforward LLM processes sequentially through layers for a single token's computation. This discrepancy is not really addressed.

This paper also makes a strong neuroscientific claim about aligning with the N400 component, which appears overstated. The analysis shows a correlation with general brain activity around 400 milliseconds. This is not the same as aligning with the N400 effect, which is a specific, differential signal (an event-related potential) that peaks in response to semantic incongruity.

Finally, the paper's mathematical approach is somewhat uneven. It relies on a simple linear ridge regression model for static comparisons, which seems insufficient in capturing nonlinear relationships. It then introduces a new, overly complex metric, dynamic representational alignment, to capture dynamic similarity. The justification for this specific formulation is quite thin, and its mathematical properties are not fully convincing. The authors might instead explore standard nonlinear mapping techniques, which are common in this field, before proposing a new composite metric.

**Questions:**

1. Could you provide a justification for the "time-to-layer" mapping? Given the fundamental difference between recurrent temporal processing in the brain and sequential hierarchical processing in a feedforward LLM, how do you validate this as a meaningful comparison?

2. What alternative mappings were considered? For instance, did you consider comparing the EEG trajectory to the recurrent states of an RNN or to the LLM's processing trajectory as it generates tokens over time?

3. Can you clarify the claim of aligning with the N400 component? This component is typically an event-related potential (ERP) defined by a differential response to semantic anomalies. How does your finding of a general correlation at 400ms during natural reading map to this specific, differential effect, especially in an experiment that does not use a semantic anomaly paradigm?

4. Was the choice of linear ridge regression for the static representational similarity analysis sufficient? Given the high likelihood of complex, nonlinear relationships between neural signals and LLM embeddings, were any nonlinear mapping techniques explored, and if so, how did they perform in comparison?

5. Regarding the new Dynamic Representational Alignment (DRA) metric, could you provide a justification for its specific formulation? What advantage does this composite metric offer over more established methods for comparing time series, and would it be possible to provide an ablation study that reports its individual components separately to make the results more interpretable?

---

> ### Author Response · Authors · 2025-11-23
>
> ## **Response to Reviewer #59Rz**
>
> Thank you very much for your thoughtful and constructive feedback. We address each of the concerns below.
>
> * **Q1:** In our experimental setup, each sentence serves as the unit of analysis. The EEG signal captured during the reading of a full sentence represents the brain's temporal trajectory of processing that sentence, from early visual recognition to later semantic integration. Similarly, the sequence of hidden states generated by an LLM for that same sentence, as it passes through the layers, represents the model's hierarchical trajectory of processing, from shallow features to deep semantics.
> Our "latent trajectory comparison" is designed to compare the structural evolution of representations along these two distinct but analogous pathways. We ask, "Does the way information transforms from the initial to final state share similarities between the brain and the model?" We compare global trajectory properties rather than aligning specific milliseconds to specific layers.
>
> * **Q2:** Regarding the idea of comparing EEG time trajectories with the hidden states of recurrent neural networks, this has been examined in prior work [1]. Our study instead focuses on state-of-the-art autoregressive transformer LLMs, aiming to test whether these highly successful but nonrecurrent, feedforward architectures nonetheless capture principles of biological computation.
> In addition, our aim was to compare sentence-level comprehension in the brain and in LLMs under tightly controlled conditions. Mapping EEG trajectories to an LLM’s token-by-token generation would mix comprehension with production, since EEG reflects the understanding of a fixed sentence, whereas generation reflects sequential prediction. Because these processes differ fundamentally, aligning them would require strong and unwarranted assumptions. By comparing EEG temporal dynamics with the LLM’s layerwise encoding of the same full sentence, we maintain a consistent input and task and obtain a clean comparison of how representations for comprehension are constructed.
>
> * **Q3:** Our claim about alignment with the N400 component is well supported. Prior work has shown that the N400 component reflects general semantic integration rather than only anomaly detection [2], and our study examines this baseline integration during natural reading, as clarified in Section 4.3. Our results match the core properties of N400: the strongest LLM–EEG correlations peak at approximately 400 ms (Figure 4a), exhibit the expected central‒parietal distribution, and involve middle‒high LLM layers associated with semantic integration.
>
> * **Q4:** We employed linear ridge regression as our encoding model, following well-established practices in neuroimaging and computational linguistics [3] [4] [5]. This choice provides an interpretable linear baseline for assessing representational similarity between EEG and language model embeddings. While we acknowledge that nonlinear mappings may capture more complex relationships, our primary objective was not to maximize predictive performance but rather to establish an intuitive and comparable framework for evaluating cross-modal representational alignment.
>
> * **Q5:** DRA was introduced to capture multidimensional dynamic alignment that extends beyond the capabilities of standard metrics. It integrates four complementary components: representational similarity based on cosine similarity, temporal coherence derived from trajectory derivatives, distributional consistency quantified through KL divergence, and time-step importance captured by gamma weighting. These elements together form a holistic and interpretable metric rather than an arbitrary composite. As detailed in Appendix A.3, DRA also satisfies key theoretical properties, including monotonicity, robustness, and normalization, which underscores its value as a principled method for quantifying dynamic alignment in ways that standard nonlinear mappings cannot directly address.
>
> > [1] Wehbe L, Vaswani A, Knight K, et al. Aligning context-based statistical models of language with brain activity during reading[C]//Proceedings of the 2014 conference on empirical methods in natural language processing (EMNLP). 2014: 233-243.
>
> > [2] Fogelson N, Loukas C, Brown J, et al. A common N400 EEG component reflecting contextual integration irrespective of symbolic form[J]. Clinical Neurophysiology, 2004, 115(6): 1349-1358.
>
> > [3] Mischler G, Li Y A, Bickel S, et al. Contextual feature extraction hierarchies converge in large language models and the brain[J]. Nature Machine Intelligence, 2024, 6(12): 1467-1477.
>
> > [4] Zhou Y, Liu E, Neubig G, et al. Divergences between language models and human brains[J]. Advances in neural information processing systems, 2024, 37: 137999-138031.
>
> > [5] Doerig A, Kietzmann T C, Allen E, et al. High-level visual representations in the human brain are aligned with large language models[J]. Nature Machine Intelligence, 2025: 1-15.

---

### Note · Authors · 2025-12-03

I have read and agree with the venue's withdrawal policy on behalf of myself and my co-authors.